# Landmark-Guided Policy Optimization for Multi-Objective Language Model Selection

**Márcio Monteiro** [1]    **Weichen Li** [1]    **Puyu Wang** [1]    **Marius Kloft** [1]    **Sophie Fellenz** [1]

## Abstract

Selecting a pretrained large language model (LLM) to fine-tune for a task-specific dataset can be time-consuming and costly. With several candidate models available to choose from, varying in size, architecture, and pretraining data, finding the best model for a specific task often involves extensive trial and error. In addition, the "best" model may not necessarily be the one with the lowest test loss, as practical considerations such as deployment costs, inference throughput, and limited search budgets might also play crucial roles. To address this, we introduce LAMPS (LAnguage Model Pareto Selection), a novel and open-source multi-objective AutoML framework that meta-learns a resource allocation policy to efficiently identify (or approximate) the Pareto front of candidate LLMs for a task-specific dataset. It is based on two key ideas: (1) landmark fine-tuning, which generates early performance indicators of the candidate models, and (2) meta-learning via reinforcement learning, which learns an effective selection policy from historical performance data (a meta-dataset). Our results show that, on held-out datasets, LAMPS reduces search time by an average of 73% compared to exhaustive search, while still covering more than 99% of the optimal target space hypervolume.

## 1. Introduction

Fine-tuning a pretrained large language model (LLM) on task-specific datasets is currently the dominant paradigm for achieving state-of-the-art performance in several natural language processing (NLP) tasks (Radford et al., 2019),

including question answering (Chowdhery et al., 2023), machine translation (Raffel et al., 2020), summarization (Aghajanyan et al., 2020), and classification (Yang, 2019). However, different pretrained models yield varying downstream performance due to differences in size, architecture, pretraining data, and other intrinsic factors. Therefore, as the set of available pretrained models is already extensive, the important question arises: How can we efficiently find the best model for a task-specific dataset?

A common practice in NLP is to select the largest available model, driven by the belief that larger models invariably provide better performance (e.g., accuracy, F1, perplexity, cross-entropy, depending on the downstream task). Although this is generally true, several studies have shown that smaller models are comparable to or even outperform larger ones for specialized tasks (Ouyang et al., 2022; Sanh et al., 2020; Hoffmann et al., 2022; Wahba et al., 2023; DeepSeek-AI et al., 2025; Wang et al., 2026). Moreover, in real-world scenarios, always choosing larger models inevitably leads to higher operational costs and greater environmental impact. This underscores the need to incorporate additional factors into the model selection process beyond a single task-specific performance metric.

A multi-objective perspective is therefore essential to capture the broader spectrum of trade-offs that practitioners face when selecting pretrained LLMs for fine-tuning. In the absence of more efficient alternatives, practitioners may turn to exhaustive search. Although theoretically sound, this method quickly becomes prohibitively expensive for a large number of candidate models, especially for target datasets with several million examples. As language models continue to expand in scale and diversity, there is an increasing need for a principled, holistic, and efficient selection strategy, especially with the growing interest in specialized LLM-based AI agents (Gutowska, 2024; Ma et al., 2024).

In this paper, we introduce **LAMPS** (**LAnguage Model Pareto Selection**), a novel and open-source multi-objective AutoML framework for selecting LLMs to fine-tune on task-specific datasets. It integrates two complementary strategies: (1) *landmark fine-tuning*, which generates early performance indicators for candidate models by evaluating them on incrementally larger subsets of the training data; and (2)

---

[1]Machine Learning Group, Department of Computer Science, RPTU University Kaiserslautern-Landau, Germany. Correspondence to: Márcio Monteiro <marcio.monteiro@cs.rptu.de>, Sophie Fellenz <sophie.fellenz@rptu.de>.

*Proceedings of the $43^{rd}$ International Conference on Machine Learning*, Seoul, South Korea. PMLR 306, 2026. Copyright 2026 by the author(s).

*meta-learning via reinforcement learning*, which leverages historical model performance data on multiple datasets to learn how to efficiently allocate training resources for new datasets. In other words, this process generates a policy that manages the selection and early stopping of candidate models, adjusting its strategy based on both observed and historical performance to efficiently discard low-potential models and prioritize promising ones.

Our main contributions are as follows: (i) Formulating the language model selection for fine-tuning explicitly as a multi-objective optimization problem; (ii) Introducing LAMPS, a novel and open-source AutoML framework combining landmark fine-tuning, meta-learning, and reinforcement learning to rapidly identify near-Pareto-optimal language models for a new task-specific dataset.

The remainder of the paper is organized as follows. Section 2 reviews relevant related work. Section 3 states the multi-objective optimization problem. The method is proposed in Section 5 and Section 6 presents the experimental setup and main findings. We conclude in Section 7.

## 2. Related Work

Selecting an appropriate base learner (model, algorithm, pipeline, etc.) for a given task has been a long-standing research topic and is usually called *model selection* (Bozdogan, 1987; Maron & Moore, 1993; McQuarrie & Tsai, 1998; Chapelle et al., 2002; Biem, 2003; Brazdil et al., 2003; Zhao & Yu, 2006; Adankon & Cheriet, 2009). Among the different approaches available, meta-learning has been a popular choice (Kalousis & Hilario, 2000; Fürnkranz et al., 2002; Brazdil & Giraud-Carrier, 2018; Jain et al., 2024; de Amorim et al., 2025; Farhadi et al., 2025), mainly due to its ability to transfer knowledge from prior learning experiences, reducing the cost of exploration and improving sample efficiency.

In this section, we provide a brief overview of the related areas that form the foundation of our LAMPS framework.

**Pretrained Model Selection in Deep Learning**  Fine-tuning pretrained deep learning models for specific downstream tasks has become the standard approach in both computer vision and natural language processing. Compared to training from scratch, fine-tuning is far more efficient and requires much less data than pretraining (Hepburn, 2018). For this reason, the ability to select the right pretrained model efficiently is becoming increasingly relevant due to the considerable computational costs and the rapid release of new models with varying sizes, architectures, training data, and capabilities. Works like You et al. (2021); Monteiro et al. (2024) explicitly address the selection of LLMs for fine-tuning, while Bayesian multi-fidelity approaches

have been proposed for pretrained-model and fine-tuning hyperparameter selection in computer vision (Arango et al., 2024). However, these methods do not consider the multi-objective aspects of model selection and have been primarily evaluated in classification tasks.

**Subsampling Landmarks**  A sampling landmark is a performance-based meta-feature, representing the performance of a particular model on samples of available data, providing a quick estimate of its performance (Brazdil et al., 2022; Pfahringer et al., 2000) and, consequently, allowing indirect characterization of the target dataset. One variant is called *subsampling landmarks*, which considers a sequence of sample sizes in increasing order, effectively representing the early stages of the learning curve (Soares et al., 2001; Fürnkranz & Petrak, 2001). This is conceptually related to the scaling laws observed in deep neural networks (Kaplan et al., 2020) and large language models (Zhang et al., 2024), which describe the predictable relationship between model performance and, among other factors, dataset size. Subsampling landmarks can thus be viewed as a localized and practical proxy for these scaling behaviors, enabling performance prediction without requiring full-scale training. Similar ideas have been applied for hyperparameter optimization (Domhan et al., 2015; Jamieson & Talwalkar, 2016; Klein et al., 2017; Li et al., 2018), which use partial learning curves to stop training poor configurations early. Such methods, however, remain inherently single-objective and cannot directly address the multi-objective settings considered in this work.

**Multi-Task and Meta-Reinforcement Learning**  Reinforcement learning is a powerful tool for sequential decision-making problems, but it often struggles with generalization to new (unknown) tasks, requiring large amounts of data to re-adapt effectively. Two areas address these limitations: multi-task reinforcement learning (MTRL) (Teh et al., 2017; Sodhani et al., 2021) and meta-reinforcement learning (Meta-RL) (Finn et al., 2017; Nichol et al., 2018; Wang et al., 2024). MTRL trains a single policy across a distribution of tasks, leveraging shared structure to improve generalization and learning efficiency. In contrast, Meta-RL focuses on learning a policy that can rapidly adapt to new tasks using limited data, typically by encoding task-specific information into its internal state or parameters. In this work, we focus on MTRL, as our goal is to evaluate policies on previously unseen datasets without further adaptation at test time.

**Hyperparameter Optimization**  Work in hyperparameter optimization (HPO), often overlapping with neural architecture search (NAS), frequently leverages early training signals to discard low-promising configurations and reduce computational cost (Falkner et al., 2018; Li et al.,

2020; Awad et al., 2021; Wistuba et al., 2022). Standard HPO methods, however, are fundamentally single-objective, and extending them to multi-objective settings typically relies on scalarization. As shown by Schmucker et al. (2021), scalarization-based adaptations usually perform significantly worse than methods explicitly designed for multi-objective search, highlighting a key limitation of conventional HPO techniques in scenarios requiring Pareto-efficient model selection. Schmucker et al. (2021) addresses this limitation by extending successive halving to the multi-objective setting using Pareto dominance, enabling more faithful approximation of the Pareto frontier. However, like most multi-objective HPO methods, it operates independently on each new task and does not exploit transferable structure across datasets. In contrast, our approach meta-learns a dataset-agnostic scheduling policy that generalizes across tasks, enabling efficient, preference-free recovery of near-Pareto-optimal solutions without per-task re-optimization.

## 3. Problem Statement

When deployment constraints are known in advance (e.g., fixed memory limit), model selection is usually posed as optimizing a single objective. In practice, however, these constraints often shift with operating conditions, available resources, or product priorities. For example, a service may favor a smaller, cheaper model during high-traffic periods and a more accurate model when demand is lower. We therefore adopt a preference-free multi-objective formulation that exposes the Pareto-efficient trade-offs, allowing model choices to be revisited without rerunning the full search.

Consider a target dataset $\mathcal{D}$ and a set $\mathcal{X}$ of candidate pre-trained language models to be fine-tuned. Then, given $n$ metrics of interest (objectives), the problem can be formulated as the following multi-objective optimization problem:

$$
\begin{aligned}
\min_{x \in \mathcal{X}} \quad & \left(f_1(x, \mathcal{D}), \ldots, f_n(x, \mathcal{D})\right) \\
\text{s.t.} \quad & f_i(x, \mathcal{D}) \leq f_i^{\max} \quad \text{for all } i = 1, \ldots, n,
\end{aligned}
\tag{1}
$$

where $f_i(x, \mathcal{D})$ represents the value of the $i$-th objective function after fine-tuning the pretrained model $x \in \mathcal{X}$ on the task-specific dataset $\mathcal{D}$, and $f_i^{\max}$ denotes an arbitrary upper bound for that objective.

Common objectives may include test loss, training cost, inference throughput, and number of model parameters. It is very common that some objectives conflict with each other. For example, achieving a lower test loss may require larger models. For this reason, there is typically no single solution that is optimal across all objectives. Hence, the notion of optimality is based on Pareto-dominance, or simply dominance, as defined below.

**Definition 3.1** (Weak dominance). A solution $x_1 \in \mathcal{X}$ weakly dominates another solution $x_2 \in \mathcal{X}$, denoted $x_1 \succeq x_2$, if $f_i(x_1, \mathcal{D}) \leq f_i(x_2, \mathcal{D})$ for all $i \in \{1, \ldots, n\}$. That is, $x_1$ is not worse than $x_2$ in all objectives.

**Definition 3.2** (Pareto-dominance). A solution $x_1 \in \mathcal{X}$ dominates another solution $x_2 \in \mathcal{X}$, denoted $x_1 \succ x_2$, if $f_i(x_1, \mathcal{D}) \leq f_i(x_2, \mathcal{D})$ for all $i \in \{1, \ldots, n\}$, with at least one of these inequalities holding strictly. That is, there is $j \in \{1, \ldots, n\}$ such that $f_j(x_1, \mathcal{D}) < f_j(x_2, \mathcal{D})$. In other words, $x_1$ dominates $x_2$ if $x_1$ is not worse than $x_2$ in all objectives, but it is better in at least one of them.

**Definition 3.3** (Pareto-optimal). A model $x^* \in \mathcal{X}$ is Pareto-optimal if there is no other $x \in \mathcal{X}$ that dominates $x^*$.

One way to evaluate and compare sets of candidate solutions is to use the *hypervolume indicator* (Guerreiro et al., 2021; Emmerich et al., 2005), which quantifies the volume of the objective space weakly dominated by a set of solutions and bounded above by a given reference point $r = [f_1^{\max}, \ldots, f_n^{\max}]^\top$. For any subset $X \subset \mathcal{X}$, the hypervolume indicator is denoted as $H_{\mathcal{D}}(X, r)$. Intuitively, each solution in $X$ defines a box in the objective space, with one corner at the objective values of the solution and the opposite corner at the reference point $r$. It is defined formally as follows:

**Definition 3.4** (Hypervolume indicator). Given a set of points $S \subset \mathbb{R}^n$ and a reference point $r \in \mathbb{R}^n$, the hypervolume indicator of $S$ is the measure of the region weakly dominated by $S$ and bounded above by $r$, i.e.,

$$
H(S, r) = \Lambda \left( \bigcup_{\substack{p \in S \\ p \leq r}} [p, r] \right),
$$

where $\Lambda(\cdot)$ denotes the Lebesgue measure and $[p, r] = \{q \in \mathbb{R}^n \mid \forall i = 1, \ldots, n : p_i \leq q_i \leq r_i\}$ denotes the box delimited below by $p \in S$ and above by $r$.

It has been shown that maximizing the hypervolume indicator is equivalent to finding the Pareto-optimal set (Guerreiro et al., 2021; Liu et al., 2019). Figure 1 illustrates this with a practical comparison, showing that the Pareto-optimal set has the highest hypervolume. Thus, the problem in (1) can be reformulated as a single-objective problem as follows:

$$
\max_{X \subset \mathcal{X}} \quad H_{\mathcal{D}}(X, r)
\tag{2}
$$

A trivial solution would involve fine-tuning all models for the target dataset (i.e., $X = \mathcal{X}$), but this is computationally expensive. To encourage computational efficiency, we introduce a regularization term $\lambda > 0$ penalizing the number of selected pretrained models:

$$
\max_{X \subset \mathcal{X}} \quad H_{\mathcal{D}}(X, r) - \lambda |X|.
\tag{3}
$$

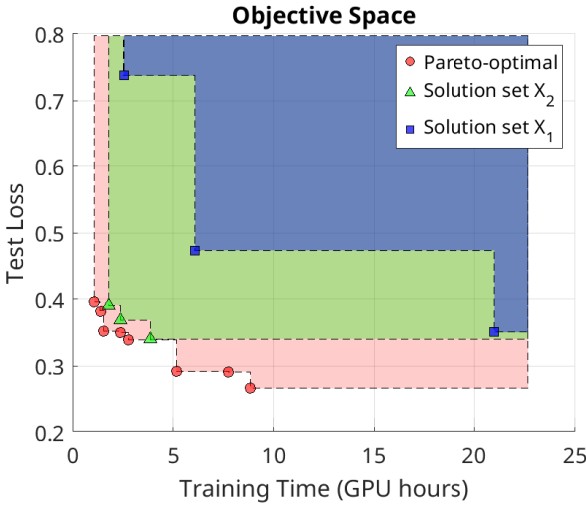

*Figure 1.* Illustration of the hypervolume indicator in a bi-objective setting, corresponding to the shaded areas. Set $X_2$ yields a larger hypervolume than $X_1$ and is closer to the true Pareto front.

## 4. Landmark Fine-tuning

Fine-tuning a pretrained model on a task-specific dataset is inevitable if one desires to evaluate its true performance and determine its suitability for a given application. However, as discussed earlier, evaluating every candidate model is computationally expensive. Prior work on hyperparameter optimization suggests that evaluating models for only a single epoch can already be a good proxy for final performance ([Egele et al., 2023](#)). However, training for just one epoch may still consume significant resources, particularly for large models and datasets.

To mitigate this inefficiency and allow for even earlier identification of unpromising candidates, we propose *landmark fine-tuning*, a multi-fidelity fine-tuning strategy based on subsampling landmarks to obtain early estimates of objective values $f_i(x, \mathcal{D})$ for $i \in \{1, \dots, n\}$.

Given that the target dataset $\mathcal{D}$ has a training and a validation split, namely $\mathcal{D}^{\text{train}}$ and $\mathcal{D}^{\text{val}}$, the core idea is to split $\mathcal{D}^{\text{train}}$ into $K$ exponentially larger subsets $\mathcal{D}_1 \dots \mathcal{D}_K$. Each subset $\mathcal{D}_k$ contains $\left\lfloor \frac{1}{2^{(K-k)}} |\mathcal{D}^{\text{train}}| \right\rfloor$ samples, where $\mathcal{D}_k \subset \mathcal{D}_{k+1}$ for $k = 1, \dots, K-1$.

The process starts by fine-tuning a pretrained model on $\mathcal{D}_1$ for a *single epoch* and evaluating it on the entire $\mathcal{D}^{\text{val}}$. Next, it continues the fine-tuning process on the subsequent (larger) subset $\mathcal{D}_2$, repeating this process up to $\mathcal{D}_K$ (100% of the training dataset). Afterward, we continue fine-tuning the model for additional epochs until convergence or another stopping criterion.

Figure 2 shows a practical example of landmark fine-tuning with $K = 8$, depicting the learning curves of five different pretrained models fine-tuned on the 20 Newsgroups dataset.

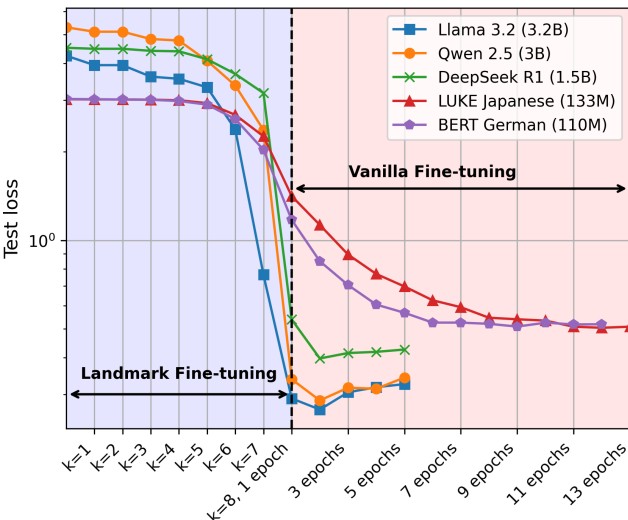

*Figure 2.* Landmark fine-tuning on the 20 Newsgroups dataset using $K = 8$. Larger models start off worse but eventually outperform smaller ones. Notably, models that improve quickly early on tend to achieve lower final loss, suggesting that the initial segments of the learning curves can help predict the overall performance.

Two non-English LLMs are included to illustrate the performance of less suitable models on an English dataset. Notice that larger models start with higher losses than smaller ones, but eventually overtake them, achieving lower final losses. In addition, among the larger models, those that improve more quickly in the initial steps tend to achieve better final loss. These observations support the idea that early segments of the training curve can indeed be predictive of final loss, with predictions becoming more accurate as additional curve segments are provided.

## 5. Meta-Learned Resource Allocation via Reinforcement Learning

Although landmark fine-tuning provides early performance estimates, it is still necessary to determine when to continue training a candidate model or not, based on partial information collected so far. To address this, we train a reinforcement learning agent on a meta-dataset of historical fine-tuning trajectories, covering a diverse set of pretrained LLMs and downstream tasks. The agent learns to allocate training resources by tracking how performance evolves across landmark steps, enabling fast and generalizable identification of near-Pareto-optimal models.

**Observation space** The observation space defines the information available to the RL agent at each decision step. At each time step $t$, the agent observes, for every candidate model, the objectives of interest (e.g., model size and validation loss), together with the number of fine-tuning epochs

that each candidate has completed.

**Action space** The action space specifies the set of decisions available to the RL agent at each step. For each time step $t$, the agent selects an action $a_t \in \{1, \ldots, m\}$, representing the index of a candidate pretrained model, where $m$ is the total number of candidates. Each action corresponds to allocating one additional fine-tuning epoch to the selected model. To improve exploration efficiency, we apply invalid action masking for completed models (Huang & Ontañón, 2022), using a binary mask that specifies which models remain available for selection. The policy then samples only from this valid subset by setting the probability of invalid actions to zero. This prevents wasted trials on completed models and makes the exploration phase more efficient, as the agent can focus its decisions on candidates that may still yield improvements.

**Termination condition** An episode corresponds to the full search process and terminates when all Pareto-optimal models have been fully fine-tuned[1], thereby achieving the maximum hypervolume. This termination condition is only necessary during policy training, where the agent has access to privileged information that indicates when the Pareto frontier has been fully explored. For rolling out the policy on new datasets (deployment), this privileged information is not necessary, and the policy simply runs until the search budget is depleted. Thanks to invalid action masking, the episode is guaranteed to terminate within a finite time, preventing the agent from getting stuck in infinite allocations to unproductive models.

**Training algorithm** For training the policy, we adopted a standard multi-task reinforcement learning setup, in which a single policy is optimized jointly across all training tasks. Each task consists of finding the Pareto-optimal models for a dataset, so we might use the terms *task* or *dataset* interchangeably. As the underlying optimizer, we adopted Proximal Policy Optimization (PPO) (Schulman et al., 2017), which provides stable on-policy updates and performs reliably in multi-task settings.

**Rewards** The reward function links our multi-objective search problem to the policy's learning process. Let $X_t \subseteq \mathcal{X}$ be the set of fully fine-tuned models by time step $t$, and let $T$ be the length of the episode. Based on (3), we could initially define a sparse reward function

$$r_t = \begin{cases} H_{\mathcal{D}}(X_t) - \lambda|X_t| & \text{if } t = T \\ 0 & \text{otherwise} \end{cases}, \qquad (4)$$

---

[1] A model is considered fully fine-tuned when its validation loss stops improving for a fixed number of consecutive epochs.

so that PPO would maximize

$$\max_{\theta} \quad \mathbb{E}_{\rho \sim \pi_\theta}\left[\sum_{t=0}^{T} \gamma^t r_t\right] = \mathbb{E}_{\rho \sim \pi_\theta}\left[H_{\mathcal{D}}(X_T) - \lambda|X_T|\right], \qquad (5)$$

where $\rho$ is a trajectory sampled using policy $\pi_\theta$, and $\gamma$ is the discount factor.

Because an episode terminates only after all Pareto-optimal models have been fully fine-tuned, $H_{\mathcal{D}}(X_T)$ is identical for every trajectory and, therefore, constant. The objective thus collapses to minimizing the expected number of models evaluated, i.e., $\mathbb{E}[-|X_T|]$. Notice that $\lambda$ also vanishes in this sparse reward setting, so we do not need to estimate it. Finally, to make the reward positive and incentivize faster convergence to the optimal, we adopted the following sparse reward function:

$$r_t = \begin{cases} \frac{|\mathcal{X} \setminus X_t|}{\Delta_t} & \text{if } t = T \\ 0 & \text{otherwise} \end{cases}, \qquad (6)$$

where $\Delta_t$ denotes the cumulative wall-clock time elapsed up to time step $t$. Intuitively, this reward favors trajectories that identify all Pareto-optimal models while minimizing unnecessary fine-tuning of dominated candidates, explicitly trading off solution completeness against time efficiency.

Importantly, this reward preserves the optimal solution of (3): among all terminating trajectories, the maximum reward is achieved if and only if $X_T = X^*$, since any trajectory that omits a Pareto-optimal model or incurs additional fine-tuning time is strictly suboptimal. Thus, the reward reshapes the learning problem without altering its optimal policies, while substantially improving learning stability for PPO. Appendix C provides additional evidence that this reward formulation consistently leads PPO to converge to the optimal solution during training.

**Meta-dataset** To meta-train a policy capable of efficiently identifying (or approximating) the Pareto-optimal set for new task-specific datasets, we conducted a fine-tuning campaign and constructed a meta-dataset containing fully recorded learning curves of 70 pretrained LLMs, each landmark fine-tuned on multiple datasets (see Appendix E). This setup enables the agent to replay arbitrary trajectories during its training, allowing the use of on-policy algorithms such as PPO. Constructing this meta-dataset (70 models, 12 datasets) required a total of 5,747 GPU-hours. Under the leave-one-out protocol, this corresponds to a mean upfront cost of 5,268 GPU-hours for 11 meta-training datasets.

**Deployment (search procedure)** Given a trained policy $\pi_\theta$ and a target dataset $\mathcal{D}$, the search procedure of LAMPS is outlined in Algorithm 1. The process begins by constructing the initial state $s_0$ through zero-shot evaluation of all candidate models on the validation split. It also serves as a sanity

**Algorithm 1** LAMPS Search Procedure

---

**Input:** training dataset $\mathcal{D}^{\text{train}}$, validation dataset $\mathcal{D}^{\text{val}}$, policy $\pi_\theta$, number of landmarks $K$
**Output:** set of non-dominated fine-tuned models $\hat{X}^\star$
Initialize time step $t \leftarrow 0$
Initialize selected model set $X \leftarrow \varnothing$
Initialize landmark level for each model $x_i$: $k_i \leftarrow 0$
Evaluate candidate models on $\mathcal{D}^{\text{val}}$ and construct initial state $s_0$
**repeat**
    Select action $a_t \leftarrow \arg\max_a \pi_\theta(a \mid s_t)$
    Update landmark level: $k_{a_t} \leftarrow \min(k_{a_t} + 1, K)$
    Fine-tune $x_{a_t}$ for one epoch on $\mathcal{D}^{\text{train}}_{k_{a_t}}$
    Evaluate updated performance on $\mathcal{D}^{\text{val}}$
    **if** stopping criterion met for model $x_{a_t}$ **then**
        $X \leftarrow X \cup \{x_{a_t}\}$
    **end if**
    Update environment state $s_{t+1}$
    $t \leftarrow t + 1$
**until** search budget is exhausted
Compute the non-dominated set $\hat{X}^\star = \{x \in X \mid \nexists y \in X : y \succ x\}$
**return** $\hat{X}^\star$

---

check to ensure that each model is available, downloaded properly, and compatible with the available hardware (and installed drivers) where the search will be performed. The policy then proceeds by selecting and executing new actions until the search budget is exhausted. In the end, dominated solutions are filtered out, so that only the best trade-offs are reported.

# 6. Experiments and Results

This section presents our experimental setup and main findings, demonstrating how well the trained policy generalizes to held-out datasets. The experiments presented in this section were conducted on eight NVIDIA A100 (40 GB) GPUs.

## 6.1. Experimental Setup

**Pretrained LLMs** We tested 70 different pretrained language models, spanning models from a few million parameters (ALBERT) to eight billion parameters (DeepSeek-R1). These models cover languages such as English, Japanese, Chinese, German, Dutch, Spanish, many of which are multilingual. The complete list of pretrained models can be found in Appendix F.

**Fine-tuning Setup** We adopted full-model fine-tuning, which updates all parameters of the pretrained models. Although parameter-efficient methods such as LoRA (Hu et al.,

2022) or layer-freezing strategies can significantly reduce computational overhead, full fine-tuning often leads to better downstream performance (Zhang et al., 2024; Shuttleworth et al., 2026). See Appendix B for details on fine-tuning hyperparameters.

**Reinforcement Learning Setup** We used the following libraries: Stable-Baselines3 (SB3) for PPO implementation and invalid action masking (Raffin et al., 2021), and the Gymnasium library for standardized environment definition (Towers et al., 2026).

**Objectives** For the optimization criteria, we focus primarily on two objectives: validation loss (measured via cross-entropy) and model size (in number of parameters). Validation loss is a widely accepted proxy for task-specific performance, and model size serves as a practical and measurable approximation of other metrics, such as VRAM, inference throughput, deployment cost, etc. These choices are not fixed for LAMPS, as the framework is objective-agnostic. Hence, any measurable objectives can be used[2], as long as the corresponding metrics are recorded in the meta-dataset. In Appendix A we show additional results for machine translation considering three objectives: validation loss, model size, and BLEU score.

**Reference point** We set the reference point by taking the worst values of the chosen objectives across the training meta-dataset and adding a 10% margin. This reference point is used only during policy training for computing the hypervolume. It is not required at test time when evaluating the trained policy (Algorithm 1).

**Baselines** We compared LAMPS with four baselines:

- **Blind**: chooses actions at random. Its performance serves as a lower bound on performance and represents the worst-case scenario.

- **Oracle**: assumes prior knowledge of the Pareto-optimal models for a given task and only allocates resources to them. The performance of this approach represents the best-case scenario. In practice, this information is not available and serves only as a theoretical upper bound.

- **ZigZag**: a simple heuristic that sorts all candidate models by their number of parameters, then selects them in an alternating order (from largest to smallest

---

[2]Choosing only highly correlated objectives collapses the Pareto frontier, effectively reducing a multi-objective problem to a single-objective one. Since adding objectives increases search complexity, it is important to select conflicting and informative objectives to make the multi-objective formulation meaningful.

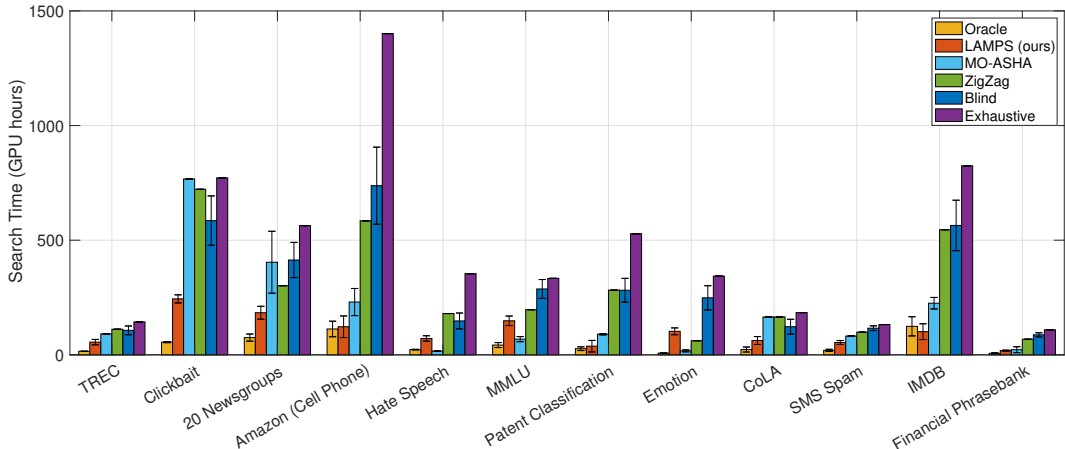

*Figure 3.* Mean time cost (in GPU-hours) to reach 99% of the optimal hypervolume indicator on held-out datasets during deployment. For reference, we also show the time to complete an exhaustive search. On average, LAMPS reduces the search time by 73.6% compared to the exhaustive search, outperforming other feasible methods in 9 out of 12 datasets.

and vice versa) in an attempt to quickly increase the covered hypervolume.

- **MO-ASHA**: Multi-objective Asynchronous Successive Halving combined with an $\epsilon$-net exploration strategy (Schmucker et al., 2021). We evaluate MO-ASHA using the same landmark fine-tuning schedule as LAMPS, ensuring a fair comparison.

**Evaluation Method** To evaluate LAMPS's generalization, we employed leave-one-out cross-validation (Hastie et al., 2009), where one dataset is held exclusively for testing. For each fold, the policy is trained on the remaining datasets for a fixed number of steps and then evaluated on the held-out dataset. This allows us to assess how well the learned policy transfers to previously unseen tasks. To ensure robustness, this procedure was repeated five times, and we report the average performance across these runs.

### 6.2. Results

To evaluate the generalization of LAMPS to unseen datasets, Figure 3 reports the deployment-time required to reach 99% of the optimal hypervolume in each held-out dataset. Recall that, in our problem formulation, achieving optimal hypervolume corresponds to identifying all Pareto-optimal models. For reference, we also include the time needed for an exhaustive search to complete. Across the twelve held-out tasks, LAMPS achieves the best performance in nine datasets (75%), whereas MO-ASHA wins only three (25%). Although MO-ASHA is the strongest baseline overall, its behavior is markedly less stable: on several datasets, its search time approaches the BLIND baseline, which never occurs with LAMPS.

To illustrate the practical implications, consider the Amazon

dataset: running an exhaustive search on a single A100 40GB NVIDIA GPU ($3.67 hourly) would cost $5,141.67, whereas LAMPS reduces the cost to just $449.58 with only a 1% degradation in the hypervolume. The strongest competing baseline, MO-ASHA, would cost $845.20 to reach the same performance.

Naturally, these per-task savings must be weighed against the initial compute investment required to construct the meta-dataset (detailed in Section 5). However, the return on investment is highly favorable: based on our leave-one-out evaluation across 12 datasets, LAMPS saves an average of 378.8 GPU-hours per held-out task. At this empirical rate, the mean upfront cost of 5,268 GPU-hours (required to meta-train on 11 datasets) is rapidly amortized after just 13.9 future tasks of comparable scale, making it exceptionally cost-effective for ongoing use.

Figure 4 provides further insight by tracking the progression of the average hypervolume over search time. For comparability, hypervolume values are normalized by the maximum hypervolume, and we report the *hypervolume loss* ($1 -$ normalized hypervolume) in logarithmic scale to highlight when the policy reaches optimality. Although LAMPS does not always reach optimality in a timely manner (compared to the other baselines), it clearly achieves near-optimal solutions quickly, eventually faster than ORACLE. This ability to deliver high-quality solutions at a fraction of the cost makes LAMPS the best trade-off between efficiency and solution quality, positioning it as a pragmatic and strong tool for practitioners.

Moreover, in multi-objective applications, the end user must ultimately select a preferred solution from the Pareto front, often revisiting trade-offs as requirements, constraints, or business priorities. By quickly providing a diverse set of

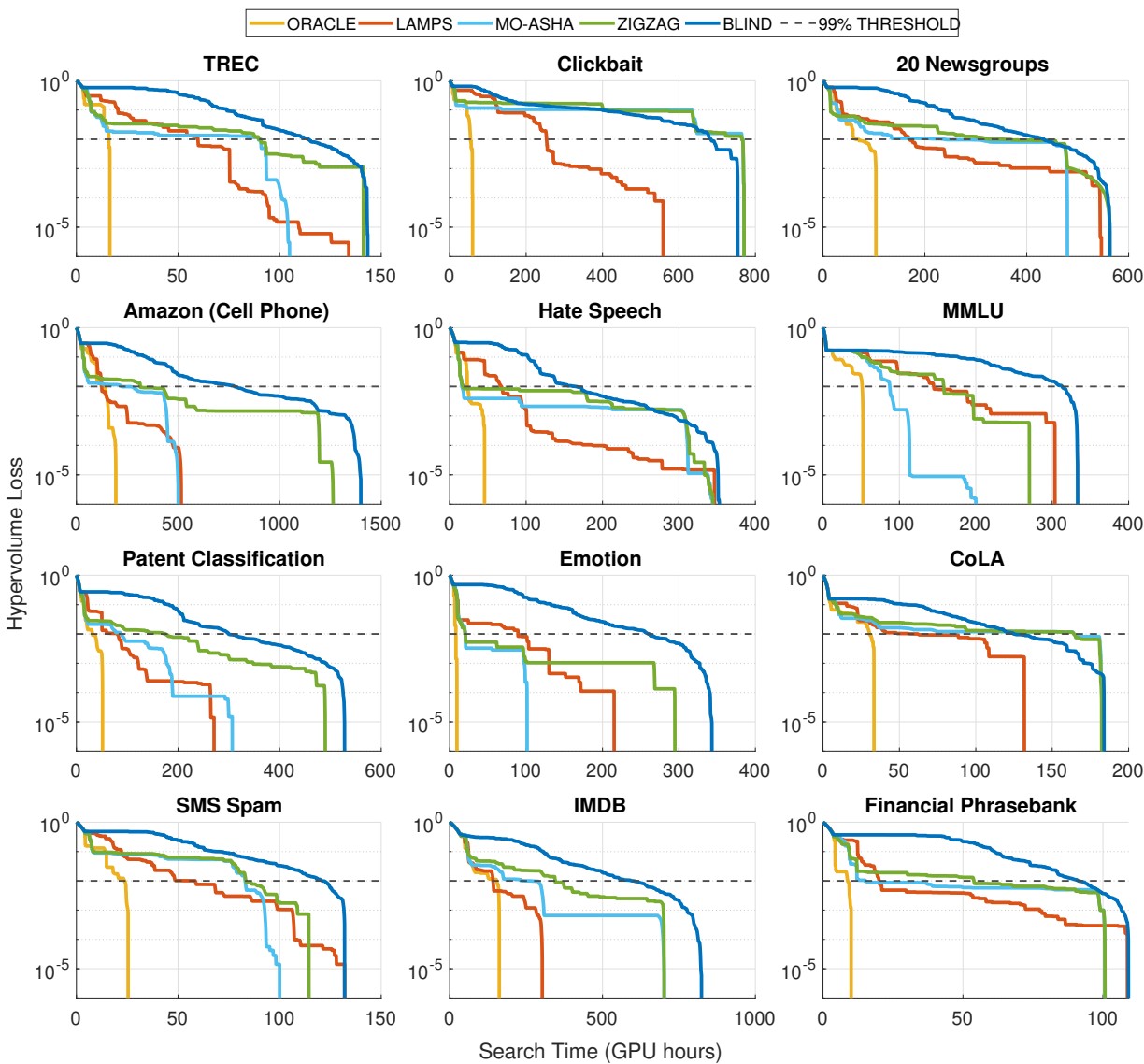

*Figure 4.* Evolution of the mean hypervolume indicator on held-out datasets as a function of search budget. LAMPS rapidly identifies near-optimal solutions (dashed line) in nine out of twelve cases, demonstrating strong generalization capabilities, even when trained on a small meta-dataset.

strong candidates, LAMPS not only accelerates the search, but also enables practitioners to reconsider or change their choice later without having to undergo another expensive search with different preferences, offering both flexibility and long-term practical value.

## 7. Conclusion

We presented LAMPS, a novel and open-source AutoML framework for efficiently selecting pretrained language models for fine-tuning, framing it as a multi-objective optimization problem. By combining landmark fine-tuning and meta-learning via reinforcement learning, LAMPS significantly

reduces search costs while maintaining near-optimal performance. Experiments show that LAMPS reduces search time by 73% on average with minimal hypervolume degradation. To our knowledge, this is the first framework that meta-learns a preference-free, dataset-agnostic scheduling policy for Pareto-efficient selection and fine-tuning for LLMs, establishing a new baseline for cost-aware AutoML and paving the way toward sustainable, high-performance deployment of foundation models.

Future work includes studying LAMPS under parameter-efficient fine-tuning regimes such as LoRA, and examining whether landmark trajectories and allocation policies remain reliable beyond the full fine-tuning setting considered here.

## Acknowledgements

The authors acknowledge support by the DFG through FOR 5359 (ID 459419731), TRR 375 (ID 511263698), SPP 2298 (IDs 441826958 and 441826958), and SPP 2331 (IDs 441958259, 553345933, and 466468799), by the Carl-Zeiss Foundation through the initiative AI-Care, and by the BMFTR award 01IS24071A. Puyu Wang acknowledges support by the Alexander-von-Humboldt Foundation through a Humboldt Research Fellowship.

The experiments were partly executed on the high performance cluster "Elwetritsch" at the RPTU Kaiserslautern-Landau which is part of the "Alliance of High Performance Computing Rheinland-Pfalz" (AHRP). We kindly acknowledge the support of RHRZ.

## Impact Statement

This paper presents work whose goal is to reduce the compute and cost required to select a pretrained language model for fine-tuning by using landmark fine-tuning signals and meta-learning a reinforcement learning policy to allocate resources efficiently. The primary positive impact is improved accessibility and sustainability: practitioners can obtain strong performance–cost trade-offs with substantially less GPU time, which can lower energy use and enable teams with less GPU capacity to run competitive model-selection pipelines. Potential risks are largely indirect and consistent with broader LLM deployment: faster selection could accelerate the deployment of models in high-stakes settings without sufficient downstream evaluation, and the choice of objectives (e.g., loss and size) may omit fairness, privacy, or safety considerations unless explicitly included. We therefore encourage users to incorporate application-relevant objectives and safeguards (e.g., robustness, bias, privacy checks) when applying LAMPS in practice.

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

*Table 1.* Time in GPU-hours to recover 99% of the optimal hypervolume on held-out datasets, optimizing for two objectives: model size and validation loss.

| Dataset | Oracle | LAMPS (ours) | MO-ASHA | ZigZag | Blind | Exhaustive |
|---|---|---|---|---|---|---|
| DE-EN | $46.0_{\pm6.0}$ | $91.1_{\pm5.9}$ | $\mathbf{50.7_{\pm5.9}}$ | 88.2 | $121.2_{\pm3.2}$ | 123.9 |
| DE-ES | $18.2_{\pm0.0}$ | $\mathbf{18.2_{\pm0.0}}$ | $35.5_{\pm5.6}$ | 53.1 | $71.3_{\pm2.4}$ | 73.6 |
| DE-FR | $18.4_{\pm0.1}$ | $\mathbf{18.3_{\pm0.1}}$ | $36.4_{\pm2.3}$ | 66.1 | $90.4_{\pm3.0}$ | 93.0 |
| DE-IT | $20.7_{\pm0.0}$ | $\mathbf{20.7_{\pm0.0}}$ | $38.0_{\pm1.8}$ | 57.7 | $76.0_{\pm1.7}$ | 77.1 |
| DE-NL | $12.4_{\pm0.0}$ | $\mathbf{12.4_{\pm0.0}}$ | $20.2_{\pm1.7}$ | 33.3 | $44.3_{\pm1.2}$ | 45.4 |
| DE-PT | $1.6_{\pm0.0}$ | $\mathbf{1.6_{\pm0.0}}$ | $3.3_{\pm0.2}$ | 4.7 | $6.1_{\pm0.1}$ | 6.2 |
| DE-RU | $2.4_{\pm1.4}$ | $15.6_{\pm0.8}$ | $\mathbf{8.4_{\pm3.0}}$ | 25.2 | $39.0_{\pm3.9}$ | 51.2 |
| EN-ES | $62.8_{\pm0.2}$ | $\mathbf{62.6_{\pm0.1}}$ | $113.6_{\pm7.6}$ | 180.0 | $239.4_{\pm7.5}$ | 245.1 |
| EN-FI | $2.6_{\pm0.5}$ | $\mathbf{3.4_{\pm0.0}}$ | $5.7_{\pm0.2}$ | 6.1 | $12.2_{\pm0.6}$ | 12.6 |
| EN-FR | $66.8_{\pm0.1}$ | $\mathbf{66.5_{\pm0.3}}$ | $92.0_{\pm20.2}$ | 233.1 | $260.1_{\pm21.4}$ | 314.0 |
| EN-IT | $22.4_{\pm0.0}$ | $\mathbf{22.3_{\pm0.0}}$ | $39.3_{\pm2.9}$ | 65.1 | $87.0_{\pm2.8}$ | 89.5 |
| EN-NL | $34.2_{\pm0.1}$ | $\mathbf{34.1_{\pm0.0}}$ | $54.3_{\pm3.4}$ | 84.1 | $110.3_{\pm2.1}$ | 111.7 |
| EN-NO | $3.0_{\pm0.0}$ | $\mathbf{2.9_{\pm0.0}}$ | $3.6_{\pm0.9}$ | 8.6 | $8.4_{\pm0.9}$ | 11.8 |
| EN-PL | $1.9_{\pm0.3}$ | $\mathbf{2.2_{\pm0.0}}$ | $4.1_{\pm0.1}$ | 6.4 | $8.7_{\pm0.3}$ | 9.0 |
| EN-PT | $2.2_{\pm0.2}$ | $\mathbf{1.5_{\pm0.0}}$ | $3.1_{\pm0.3}$ | 4.5 | $5.6_{\pm0.2}$ | 5.8 |
| EN-RU | $2.7_{\pm0.9}$ | $13.2_{\pm0.6}$ | $\mathbf{5.5_{\pm2.4}}$ | 23.8 | $30.0_{\pm5.1}$ | 47.8 |
| EN-SV | $3.2_{\pm0.6}$ | $\mathbf{3.3_{\pm0.0}}$ | $4.0_{\pm0.7}$ | 8.6 | $8.7_{\pm0.9}$ | 11.5 |
| ES-FI | $3.4_{\pm0.5}$ | $\mathbf{3.6_{\pm0.0}}$ | $6.2_{\pm0.6}$ | 9.1 | $11.9_{\pm0.3}$ | 12.2 |
| ES-FR | $32.3_{\pm0.1}$ | $\mathbf{32.2_{\pm0.1}}$ | $53.6_{\pm4.8}$ | 103.1 | $116.7_{\pm7.8}$ | 144.0 |
| ES-IT | $24.0_{\pm0.0}$ | $\mathbf{23.9_{\pm0.0}}$ | $41.5_{\pm2.9}$ | 62.6 | $83.4_{\pm2.0}$ | 85.0 |
| ES-NL | $28.7_{\pm0.0}$ | $\mathbf{28.7_{\pm0.0}}$ | $45.7_{\pm3.4}$ | 73.2 | $96.5_{\pm2.1}$ | 97.8 |
| ES-NO | $3.4_{\pm0.0}$ | $\mathbf{3.4_{\pm0.0}}$ | $3.9_{\pm0.7}$ | 9.1 | $8.8_{\pm0.9}$ | 12.4 |
| ES-PT | $1.9_{\pm0.0}$ | $\mathbf{1.5_{\pm0.0}}$ | $3.0_{\pm0.2}$ | 4.5 | $6.2_{\pm0.1}$ | 6.3 |
| ES-RU | $3.6_{\pm1.2}$ | $15.7_{\pm0.0}$ | $\mathbf{3.7_{\pm2.2}}$ | 25.9 | $36.3_{\pm4.5}$ | 50.5 |
| FI-FR | $3.6_{\pm0.3}$ | $\mathbf{6.0_{\pm0.3}}$ | $10.0_{\pm0.3}$ | 11.2 | $10.9_{\pm0.4}$ | 11.2 |
| FI-NO | $4.8_{\pm0.3}$ | $\mathbf{3.5_{\pm0.0}}$ | $6.1_{\pm0.5}$ | 9.1 | $11.4_{\pm0.5}$ | 12.1 |
| FI-PL | $2.8_{\pm0.0}$ | $\mathbf{2.7_{\pm0.0}}$ | $4.8_{\pm0.1}$ | 7.8 | $10.3_{\pm0.4}$ | 10.6 |
| FR-IT | $11.6_{\pm0.0}$ | $\mathbf{11.5_{\pm0.0}}$ | $20.5_{\pm1.4}$ | 32.6 | $43.0_{\pm0.9}$ | 43.7 |
| FR-NL | $30.5_{\pm0.0}$ | $\mathbf{30.5_{\pm0.0}}$ | $49.2_{\pm1.0}$ | 82.9 | $111.5_{\pm3.9}$ | 114.8 |
| FR-NO | $3.1_{\pm0.4}$ | $\mathbf{3.3_{\pm0.0}}$ | $5.3_{\pm0.4}$ | 8.6 | $10.5_{\pm0.4}$ | 11.4 |
| FR-PL | $2.4_{\pm0.3}$ | $\mathbf{2.6_{\pm0.0}}$ | $4.2_{\pm1.0}$ | 6.8 | $9.3_{\pm0.3}$ | 9.5 |
| FR-PT | $1.5_{\pm0.0}$ | $\mathbf{1.5_{\pm0.0}}$ | $2.9_{\pm0.1}$ | 4.7 | $6.2_{\pm0.1}$ | 6.4 |
| FR-RU | $1.6_{\pm0.5}$ | $7.5_{\pm0.4}$ | $\mathbf{3.9_{\pm0.7}}$ | 12.5 | $17.8_{\pm2.3}$ | 25.0 |
| FR-SV | $2.9_{\pm0.4}$ | $\mathbf{3.1_{\pm0.0}}$ | $3.8_{\pm1.2}$ | 8.5 | $8.4_{\pm0.6}$ | 11.4 |
| IT-NL | $2.3_{\pm0.0}$ | $\mathbf{2.3_{\pm0.0}}$ | $4.2_{\pm0.2}$ | 6.2 | $8.7_{\pm0.2}$ | 8.9 |
| IT-PT | $1.8_{\pm0.0}$ | $\mathbf{1.4_{\pm0.0}}$ | $2.9_{\pm0.2}$ | 4.7 | $5.8_{\pm0.1}$ | 5.9 |
| IT-RU | $4.2_{\pm0.0}$ | $16.2_{\pm0.0}$ | $\mathbf{4.3_{\pm1.6}}$ | 27.2 | $37.6_{\pm5.1}$ | 54.4 |
| IT-SV | $3.3_{\pm0.2}$ | $\mathbf{3.3_{\pm0.0}}$ | $4.7_{\pm0.9}$ | 8.7 | $9.1_{\pm0.9}$ | 11.5 |

# A. Additional Experiments: Machine Translation

To further assess the generality, objective-agnosticism, and scalability of LAMPS, we conducted additional experiments in the domain of machine translation. Using 4x NVIDIA A100 (40GB) GPUs, we constructed a meta-dataset comprising 38 translation directions from the OPUS Books corpus, a collection of copyright-free literary texts spanning a wide range of languages (Tiedemann, 2012).

## A.1. Two Objectives

Table 1 reports the time (in GPU-hours) to recover 99% of the optimal hypervolume when optimizing for model size and validation loss. The results show that LAMPS transfers meta-learned knowledge effectively to the majority of held-out task-specific datasets, being comparable to the ORACLE in 32 out of 38 cases.

## A.2. Three Objectives

To evaluate how well LAMPS scales to higher-dimensional objective spaces, we extend our analysis to a three-objective setting involving model size, validation loss, and BLEU score. As shown in Table 2, the ORACLE requires substantially

*Table 2.* Time in GPU-hours to recover 99% of the optimal hypervolume on held-out datasets, optimizing for three objectives: model size, validation loss and BLEU score.

| Dataset | Oracle | LAMPS (ours) | MO-ASHA | ZigZag | Blind | Exhaustive |
|---|---|---|---|---|---|---|
| DE-EN | $63.0_{\pm 3.7}$ | $88.9_{\pm 1.1}$ | $94.5_{\pm 4.5}$ | **88.4** | $120.6_{\pm 3.3}$ | 123.9 |
| DE-ES | $30.6_{\pm 5.1}$ | $51.4_{\pm 0.6}$ | $\mathbf{49.0_{\pm 6.9}}$ | 53.1 | $71.4_{\pm 2.6}$ | 73.6 |
| DE-FR | $26.0_{\pm 2.7}$ | $62.7_{\pm 3.2}$ | $\mathbf{34.5_{\pm 5.6}}$ | 66.1 | $90.4_{\pm 2.8}$ | 93.0 |
| DE-IT | $33.9_{\pm 1.9}$ | $\mathbf{51.8_{\pm 1.1}}$ | $54.6_{\pm 13.3}$ | 57.7 | $76.5_{\pm 0.9}$ | 77.1 |
| DE-NL | $29.0_{\pm 2.5}$ | $\mathbf{32.1_{\pm 0.6}}$ | $44.0_{\pm 0.8}$ | 45.3 | $44.9_{\pm 0.7}$ | 45.4 |
| DE-PT | $3.7_{\pm 0.1}$ | $\mathbf{4.2_{\pm 0.1}}$ | $6.2_{\pm 0.0}$ | 6.2 | $6.2_{\pm 0.1}$ | 6.2 |
| DE-RU | $13.3_{\pm 1.4}$ | $36.1_{\pm 0.1}$ | $\mathbf{17.3_{\pm 2.4}}$ | 29.8 | $49.9_{\pm 1.4}$ | 51.2 |
| EN-ES | $62.9_{\pm 0.1}$ | $\mathbf{94.6_{\pm 35.2}}$ | $99.3_{\pm 8.8}$ | 180.1 | $241.7_{\pm 4.2}$ | 245.1 |
| EN-FI | $7.3_{\pm 0.3}$ | $\mathbf{8.3_{\pm 0.1}}$ | $11.9_{\pm 0.2}$ | 12.6 | $12.2_{\pm 0.3}$ | 12.6 |
| EN-FR | $77.6_{\pm 3.1}$ | $222.4_{\pm 0.0}$ | $\mathbf{126.5_{\pm 8.9}}$ | 240.1 | $309.1_{\pm 6.3}$ | 314.0 |
| EN-IT | $23.9_{\pm 0.0}$ | $62.7_{\pm 2.5}$ | $\mathbf{39.5_{\pm 2.2}}$ | 65.1 | $87.0_{\pm 2.5}$ | 89.5 |
| EN-NL | $61.4_{\pm 3.0}$ | $\mathbf{81.9_{\pm 1.3}}$ | $104.0_{\pm 4.7}$ | 84.1 | $109.6_{\pm 2.2}$ | 111.7 |
| EN-NO | $5.8_{\pm 0.4}$ | $\mathbf{8.9_{\pm 0.5}}$ | $11.2_{\pm 0.1}$ | 11.8 | $11.5_{\pm 0.3}$ | 11.8 |
| EN-PL | $4.7_{\pm 0.0}$ | $\mathbf{5.8_{\pm 0.4}}$ | $8.7_{\pm 0.3}$ | 9.0 | $8.7_{\pm 0.3}$ | 9.0 |
| EN-PT | $3.2_{\pm 0.0}$ | $\mathbf{3.6_{\pm 0.0}}$ | $5.3_{\pm 1.0}$ | 4.5 | $5.7_{\pm 0.1}$ | 5.8 |
| EN-RU | $10.6_{\pm 0.1}$ | $28.4_{\pm 7.4}$ | $\mathbf{17.7_{\pm 2.9}}$ | 23.8 | $46.4_{\pm 1.5}$ | 47.8 |
| EN-SV | $5.2_{\pm 0.3}$ | $\mathbf{7.7_{\pm 0.0}}$ | $11.0_{\pm 0.1}$ | 11.5 | $10.2_{\pm 0.6}$ | 11.5 |
| ES-FI | $6.8_{\pm 0.8}$ | $\mathbf{8.8_{\pm 0.3}}$ | $11.6_{\pm 0.0}$ | 12.2 | $11.6_{\pm 0.4}$ | 12.2 |
| ES-FR | $86.5_{\pm 1.2}$ | $\mathbf{99.4_{\pm 2.0}}$ | $143.8_{\pm 0.0}$ | 143.7 | $142.6_{\pm 2.2}$ | 144.0 |
| ES-IT | $43.6_{\pm 0.7}$ | $\mathbf{59.1_{\pm 1.8}}$ | $81.8_{\pm 1.6}$ | 64.3 | $84.4_{\pm 0.9}$ | 85.0 |
| ES-NL | $66.5_{\pm 2.2}$ | $\mathbf{81.9_{\pm 4.9}}$ | $97.7_{\pm 0.0}$ | 97.7 | $96.9_{\pm 1.3}$ | 97.8 |
| ES-NO | $6.7_{\pm 0.7}$ | $\mathbf{7.5_{\pm 0.3}}$ | $12.4_{\pm 0.0}$ | 12.4 | $11.9_{\pm 0.5}$ | 12.4 |
| ES-PT | $3.9_{\pm 0.1}$ | $\mathbf{3.8_{\pm 0.2}}$ | $6.3_{\pm 0.0}$ | 6.3 | $6.3_{\pm 0.1}$ | 6.3 |
| ES-RU | $12.5_{\pm 3.4}$ | $34.5_{\pm 1.1}$ | $\mathbf{19.0_{\pm 0.9}}$ | 25.9 | $48.8_{\pm 1.4}$ | 50.5 |
| FI-FR | $5.1_{\pm 0.0}$ | $\mathbf{7.0_{\pm 0.5}}$ | $10.6_{\pm 0.1}$ | 11.2 | $10.9_{\pm 0.3}$ | 11.2 |
| FI-NO | $8.6_{\pm 0.3}$ | $\mathbf{8.4_{\pm 0.1}}$ | $12.1_{\pm 0.0}$ | 12.1 | $11.9_{\pm 0.3}$ | 12.1 |
| FI-PL | $6.6_{\pm 0.2}$ | $\mathbf{7.1_{\pm 0.4}}$ | $10.6_{\pm 0.0}$ | 10.6 | $10.3_{\pm 0.4}$ | 10.6 |
| FR-IT | $18.7_{\pm 0.0}$ | $30.5_{\pm 0.3}$ | $\mathbf{20.1_{\pm 1.5}}$ | 32.6 | $42.9_{\pm 1.1}$ | 43.7 |
| FR-NL | $74.3_{\pm 5.2}$ | $\mathbf{80.2_{\pm 1.0}}$ | $114.8_{\pm 0.0}$ | 114.6 | $113.9_{\pm 1.6}$ | 114.8 |
| FR-NO | $6.2_{\pm 0.7}$ | $\mathbf{6.7_{\pm 0.4}}$ | $11.4_{\pm 0.0}$ | 11.4 | $11.2_{\pm 0.2}$ | 11.4 |
| FR-PL | $5.6_{\pm 0.2}$ | $\mathbf{5.9_{\pm 0.0}}$ | $9.3_{\pm 0.2}$ | 9.5 | $9.3_{\pm 0.2}$ | 9.5 |
| FR-PT | $3.6_{\pm 0.2}$ | $\mathbf{4.3_{\pm 0.1}}$ | $6.4_{\pm 0.0}$ | 6.4 | $6.3_{\pm 0.1}$ | 6.4 |
| FR-RU | $7.3_{\pm 0.1}$ | $13.3_{\pm 3.1}$ | $\mathbf{10.8_{\pm 0.2}}$ | 18.6 | $24.5_{\pm 0.7}$ | 25.0 |
| FR-SV | $7.2_{\pm 0.4}$ | $\mathbf{7.7_{\pm 0.1}}$ | $11.4_{\pm 0.0}$ | 11.4 | $11.1_{\pm 0.3}$ | 11.4 |
| IT-NL | $6.1_{\pm 0.2}$ | $\mathbf{6.6_{\pm 0.5}}$ | $8.9_{\pm 0.0}$ | 8.9 | $8.8_{\pm 0.1}$ | 8.9 |
| IT-PT | $3.8_{\pm 0.1}$ | $\mathbf{4.2_{\pm 0.1}}$ | $5.7_{\pm 0.3}$ | 5.9 | $5.9_{\pm 0.0}$ | 5.9 |
| IT-RU | $13.0_{\pm 1.5}$ | $35.3_{\pm 1.8}$ | $\mathbf{11.6_{\pm 2.7}}$ | 27.2 | $50.6_{\pm 2.6}$ | 54.4 |
| IT-SV | $6.9_{\pm 0.4}$ | $\mathbf{7.6_{\pm 0.1}}$ | $11.5_{\pm 0.0}$ | 11.5 | $11.2_{\pm 0.4}$ | 11.5 |

more time to recover 99% of the optimal hypervolume than in the 2D case. This increase reflects the expansion of the Pareto frontier (now a surface) when BLEU is added, making the search space more challenging to find or approximate.

Despite this increased complexity, LAMPS remains the strongest overall method by a large margin, achieving the best search performance on 27 of 38 datasets (71%). Although MO-ASHA becomes more competitive in this 3D setting, increasing its win rate to 10/38 (26%), its performance also becomes significantly less stable: on several language pairs, it drops to the level of the BLIND baseline, which has not been observed in any of the 2D experiments. This widening performance gap indicates that LAMPS scales more reliably and consistently as the dimensionality of the objective space increases.

### A.3. Empirical Analysis of Failure Modes

Although LAMPS demonstrates efficient recovery of the Pareto-optimal set for the majority of datasets, Table 1 reveals a consistent degradation in search efficiency on specific held-out tasks: across all translation directions involving Russian (e.g., ES-RU, DE-RU, EN-RU, IT-RU, FR-RU) and German to English (DE-EN).

To investigate these failures, we empirically measured pairwise dataset similarity by training the policy on a single dataset

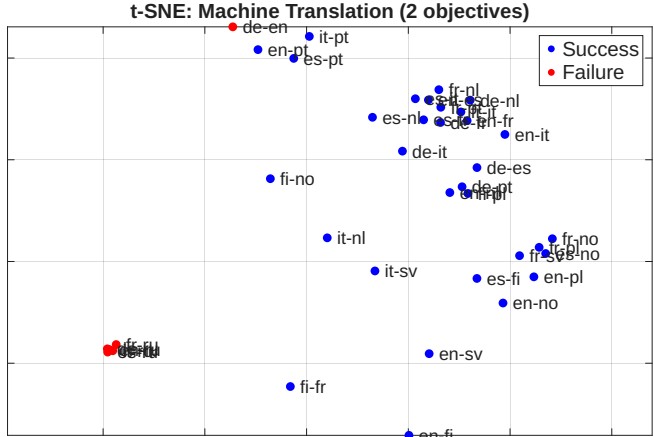

*Figure 5.* 2D t-SNE visualization of task similarities based on transfer rewards. Tasks where LAMPS underperforms ("Failure", notably Russian translations) form isolated clusters, illustrating that deviation from the majority meta-training distribution correlates with reduced search efficiency.

and evaluating it on all others, using the transfer reward as a proxy for source-target similarity. As illustrated in the 2D t-SNE visualization of this transfer matrix (Figure 5), the Russian translation datasets form a tight cluster that is highly isolated from the rest of the others datasets. A plausible explanation is that the policy implicitly favors the underlying dynamics of the majority distribution seen during training. When searching on a target dataset that falls outside this dense cluster, the policy struggles to adapt, leading to less directed search behavior and less efficient resource allocation.

## B. Hyperparameters

### B.1. Fine-Tuning

We used the Trainer module from Hugging Face's **transformers** library for fine-tuning. The key hyperparameters and settings were as follows:

- Optimizer: AdamW

- Learning rate: $7 \times 10^{-6}$

- Batch size: Automatically determined based on available hardware

- Early stopping patience: 3 epochs

- Mixed precision: Enabled (BF16)

- Number of landmark subsets ($K$): 8

All unspecified settings follow the default values defined in Trainer module.

### B.2. PPO

For the PPO algorithm, we used the implementation from Stable Baselines3 library. The key hyperparameters and settings were as follows:

- Learning rate: $1 \times 10^{-4}$

- Minibatch size: 256

- Num. epochs: 15

- Discount ($\gamma$): 0.99

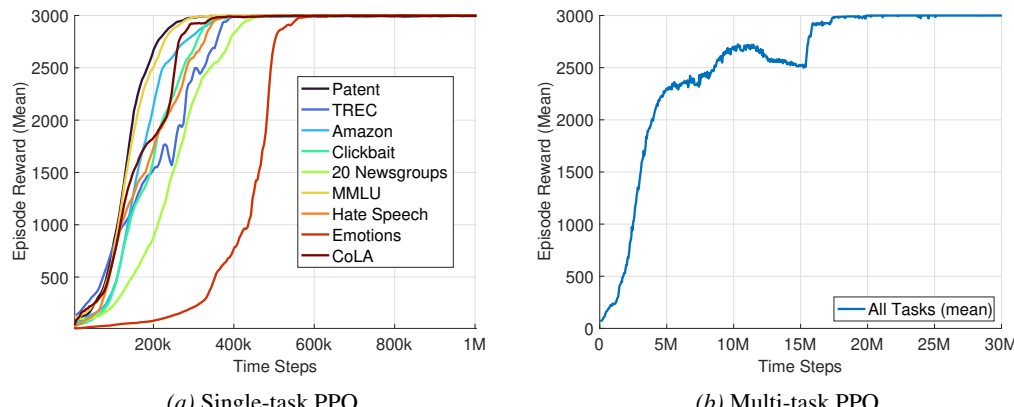

*(a)* Single-task PPO.  *(b)* Multi-task PPO.

*Figure 6.* Normalized reward progression during policy training with PPO. Multi-task RL converges more slowly but reaches the same optimal reward as single-task training.

- GAE parameter ($\lambda$): 0.97

- Clip range: 0.20

- VF coeff. $c_1$: 0.5

- Entropy coeff. $c_2$: 0.23

All policies were trained using the Gymnasium environment API with invalid action masking.

## C. Reward Optimality

This section provides both a formal justification and empirical evidence for the reward formulation introduced in (6). We first show that the proposed terminal reward preserves the optimal solutions of the original multi-objective selection problem, and then empirically demonstrate that it leads to stable convergence when optimized with PPO.

### C.1. Reward Optimality

**Lemma C.1** (Policy-optimality of the terminal reward). *Assume that, during policy training, an episode terminates only when all Pareto-optimal models $\mathcal{X}^*$ have been fully fine-tuned, i.e., $\mathcal{X}^* \subseteq X_T$. Let the terminal reward be defined as*

$$r_T \;=\; \frac{|\mathcal{X} \setminus X_T|}{\Delta_T}, \qquad \Delta_T > 0,$$

*with $r_t = 0$ for all $t < T$. Then any reward-maximizing terminating trajectory satisfies $X_T = \mathcal{X}^*$. Moreover, among trajectories with $X_T = \mathcal{X}^*$, maximizing $r_T$ is equivalent to minimizing the total wall-clock time $\Delta_T$.*

*Proof.* By the termination assumption, every terminating trajectory satisfies $\mathcal{X}^* \subseteq X_T$. If $X_T \neq \mathcal{X}^*$, then $X_T$ strictly contains at least one dominated model, implying

$$|\mathcal{X} \setminus X_T| < |\mathcal{X} \setminus \mathcal{X}^*|.$$

For any fixed $\Delta_T > 0$, this yields a strictly smaller reward than that obtained with $X_T = \mathcal{X}^*$. Hence, no trajectory with $X_T \supsetneq \mathcal{X}^*$ can be reward-maximizing. Restricting to trajectories with $X_T = \mathcal{X}^*$, the numerator of $r_T$ is constant, so maximizing the reward is equivalent to minimizing $\Delta_T$. □

*Remark* C.2. Lemma C.1 shows that the proposed terminal reward constitutes policy-invariant reward shaping under the training-time termination condition, preserving the optimal solutions of the original multi-objective selection objective while explicitly favoring time-efficient discovery of the Pareto-optimal set.

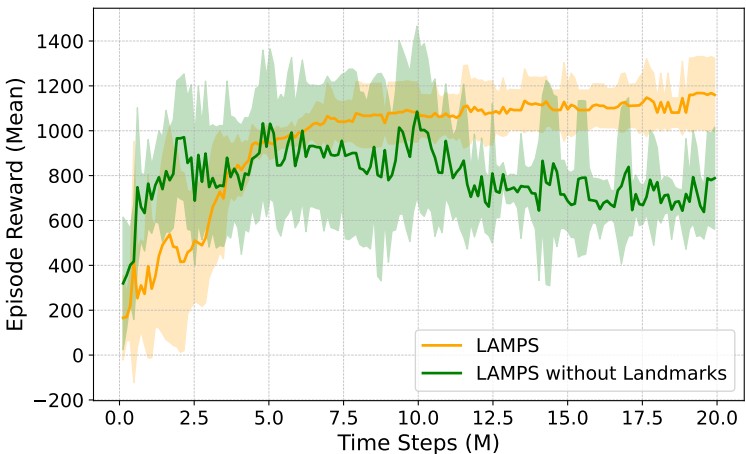

*Figure 7.* Ablation study removing landmark fine-tuning. Mean episode reward as a function of training steps for the full method (LAMPS) and a variant without landmark fine-tuning. Removing landmark fine-tuning leads to consistently lower rewards and slower convergence, highlighting its role in stabilizing and accelerating learning.

*Remark* C.3. While step-wise rewards provide more frequent feedback, defining them to accurately reflect progress toward a global, multi-objective Pareto frontier is non-trivial. Instead, we rely on a theoretically grounded sparse terminal reward. As shown in Lemma C.1, this formulation constitutes policy-invariant reward shaping under the training-time termination condition, preserving the optimal solutions of the original problem while explicitly penalizing unnecessary fine-tuning.

### C.2. Empirical Convergence Analysis

To complement the theoretical result above, we provide empirical evidence that the reward in Eq. (6) effectively guides policy optimization toward the optimal solution set. Figure 6 shows a representative evolution of the normalized reward during PPO training, for both single-task and multi-task reinforcement learning (MTRL).

For interpretability, rewards are normalized such that a value of 3000 corresponds to the optimal reward achieved when the policy exclusively fine-tunes Pareto-optimal models and reaches the maximal hypervolume in minimal time. In both settings, the learned policy exhibits a consistent upward trend and converges to the optimal reward. As expected, multi-task training requires more iterations to converge due to the increased diversity of tasks, but ultimately reaches the same optimal reward level, confirming that the proposed reward supports stable learning and effective generalization across datasets.

## D. Ablation Studies

### D.1. Removing Landmark Fine-tuning

Removing the landmark fine-tuning component reduces LAMPS to a purely black-box optimization method, whereas the full approach can be viewed as a gray-box optimizer that leverages additional structural information.

Figure 7 presents an ablation study that evaluates the impact of landmark fine-tuning. Results are averaged over five independent training runs, with shaded regions indicating the standard deviation. Removing this component consistently degrades performance throughout training, resulting in slower convergence and lower final rewards. This indicates that landmark fine-tuning, in fact, improves learning stability and efficiency.

### D.2. Single Objective

Although our primary focus is multi-objective optimization, LAMPS applies directly to the single-objective case without modification. We therefore evaluate it on the same machine translation benchmark, optimizing only validation loss and measuring the time required to reach 99% of the optimal value.

In this setting, LAMPS achieves a win rate of 35/38 datasets (92.1%), exhibiting even stronger relative performance than

*Table 3.* Time in GPU-hours to recover 99% of the optimal hypervolume on held-out datasets for single objective scenario.

| Dataset | Oracle | LAMPS (ours) | SHA | Hyperband | BOHB | ZigZag | Random | Exhaustive |
|---|---|---|---|---|---|---|---|---|
| DE-EN | 5.5 | $\mathbf{5.5_{\pm 0.0}}$ | **4.4** | $29.2_{\pm 10.6}$ | 20.0 | 11.7 | $37.4_{\pm 11.0}$ | 123.9 |
| DE-ES | 3.3 | $\mathbf{3.3_{\pm 0.0}}$ | 4.1 | $24.2_{\pm 11.2}$ | 16.3 | 20.5 | $31.3_{\pm 9.5}$ | 73.6 |
| DE-FR | 4.3 | $\mathbf{4.3_{\pm 0.0}}$ | 5.2 | $19.5_{\pm 1.8}$ | 20.0 | 9.5 | $35.1_{\pm 8.7}$ | 93.0 |
| DE-IT | 4.1 | $\mathbf{4.1_{\pm 0.0}}$ | 4.9 | $20.0_{\pm 5.1}$ | 16.9 | 21.6 | $35.7_{\pm 9.7}$ | 77.1 |
| DE-NL | 2.6 | $\mathbf{2.6_{\pm 0.0}}$ | 3.0 | $11.6_{\pm 2.8}$ | 9.9 | 12.6 | $21.7_{\pm 5.5}$ | 45.4 |
| DE-PT | 0.5 | $\mathbf{0.5_{\pm 0.0}}$ | 0.6 | $2.5_{\pm 1.0}$ | 1.9 | 2.0 | $3.5_{\pm 0.6}$ | 6.2 |
| DE-RU | 0.5 | $4.7_{\pm 0.0}$ | - | $18.7_{\pm 10.8}$ | 9.4 | 25.2 | $21.2_{\pm 6.1}$ | 51.2 |
| EN-ES | 10.7 | $\mathbf{10.7_{\pm 0.0}}$ | 13.2 | $59.7_{\pm 9.8}$ | 52.7 | 39.2 | $97.7_{\pm 28.2}$ | 245.1 |
| EN-FI | 0.7 | $\mathbf{0.9_{\pm 0.0}}$ | 1.0 | $3.2_{\pm 0.2}$ | 2.9 | 1.1 | $5.3_{\pm 1.2}$ | 12.6 |
| EN-FR | 14.8 | $\mathbf{14.8_{\pm 0.0}}$ | 18.0 | $93.2_{\pm 51.4}$ | 76.1 | 89.4 | $158.8_{\pm 38.9}$ | 314.0 |
| EN-IT | 4.4 | $\mathbf{4.4_{\pm 0.0}}$ | 5.3 | $50.3_{\pm 26.9}$ | 21.2 | 24.5 | $43.4_{\pm 12.8}$ | 89.5 |
| EN-NL | 5.2 | $\mathbf{5.2_{\pm 0.0}}$ | 6.2 | $27.7_{\pm 10.6}$ | 22.5 | 28.7 | $46.4_{\pm 11.9}$ | 111.7 |
| EN-NO | 0.6 | $\mathbf{0.6_{\pm 0.0}}$ | 0.8 | $2.8_{\pm 0.3}$ | 2.8 | 2.1 | $5.7_{\pm 1.2}$ | 11.8 |
| EN-PL | 0.6 | $\mathbf{0.6_{\pm 0.0}}$ | 0.7 | $4.4_{\pm 1.8}$ | 2.4 | 2.9 | $4.2_{\pm 1.1}$ | 9.0 |
| EN-PT | 0.5 | $\mathbf{0.5_{\pm 0.0}}$ | 0.6 | $2.1_{\pm 0.3}$ | 2.0 | 2.1 | $3.2_{\pm 0.5}$ | 5.8 |
| EN-RU | 0.5 | $3.9_{\pm 0.9}$ | - | $10.4_{\pm 2.2}$ | 8.7 | 23.8 | $23.7_{\pm 6.8}$ | 47.8 |
| EN-SV | 0.6 | $\mathbf{0.7_{\pm 0.0}}$ | 0.9 | $3.8_{\pm 1.6}$ | 2.9 | 1.1 | $4.4_{\pm 1.2}$ | 11.5 |
| ES-FI | 0.8 | $\mathbf{0.8_{\pm 0.0}}$ | 1.0 | $3.1_{\pm 0.6}$ | 3.0 | 1.2 | $4.9_{\pm 1.1}$ | 12.2 |
| ES-FR | 6.1 | $\mathbf{6.1_{\pm 0.0}}$ | 7.6 | $30.4_{\pm 1.5}$ | 31.6 | 15.6 | $45.8_{\pm 16.2}$ | 144.0 |
| ES-IT | 4.1 | $\mathbf{4.1_{\pm 0.0}}$ | 5.0 | $20.0_{\pm 4.4}$ | 19.7 | 16.4 | $35.0_{\pm 8.3}$ | 85.0 |
| ES-NL | 5.2 | $\mathbf{5.2_{\pm 0.0}}$ | 6.1 | $38.4_{\pm 19.5}$ | 21.8 | 29.0 | $47.9_{\pm 11.8}$ | 97.8 |
| ES-NO | 0.7 | $\mathbf{0.7_{\pm 0.0}}$ | 0.9 | $4.8_{\pm 1.8}$ | 3.1 | 3.7 | $6.4_{\pm 1.6}$ | 12.4 |
| ES-PT | 0.4 | $\mathbf{0.4_{\pm 0.0}}$ | 0.6 | $2.1_{\pm 0.4}$ | 1.9 | 2.1 | $3.1_{\pm 0.6}$ | 6.3 |
| ES-RU | 0.5 | $5.6_{\pm 0.8}$ | **1.0** | $22.3_{\pm 14.4}$ | 8.0 | 25.9 | $24.0_{\pm 6.1}$ | 50.5 |
| FI-FR | 0.7 | $\mathbf{1.3_{\pm 0.0}}$ | - | $3.2_{\pm 0.6}$ | 2.9 | 1.4 | $6.1_{\pm 1.6}$ | 11.2 |
| FI-NO | 0.6 | $\mathbf{0.8_{\pm 0.0}}$ | 1.2 | $3.1_{\pm 0.6}$ | 2.9 | 1.3 | $4.5_{\pm 1.3}$ | 12.1 |
| FI-PL | 0.6 | $\mathbf{0.6_{\pm 0.0}}$ | 0.8 | $3.0_{\pm 0.6}$ | 2.4 | 3.3 | $4.7_{\pm 1.4}$ | 10.6 |
| FR-IT | 2.2 | $\mathbf{2.2_{\pm 0.0}}$ | 2.6 | $10.3_{\pm 2.7}$ | 10.4 | 8.5 | $19.7_{\pm 5.3}$ | 43.7 |
| FR-NL | 5.1 | $\mathbf{5.1_{\pm 0.0}}$ | 6.1 | $50.5_{\pm 27.5}$ | 25.0 | 32.5 | $51.6_{\pm 13.6}$ | 114.8 |
| FR-NO | 0.6 | $\mathbf{0.6_{\pm 0.0}}$ | 0.8 | $3.7_{\pm 0.7}$ | 2.8 | 2.2 | $5.3_{\pm 1.2}$ | 11.4 |
| FR-PL | 0.5 | $\mathbf{0.5_{\pm 0.0}}$ | 0.7 | $3.9_{\pm 2.4}$ | 2.4 | 3.1 | $4.7_{\pm 1.3}$ | 9.5 |
| FR-PT | 0.4 | $\mathbf{0.4_{\pm 0.0}}$ | 0.5 | $3.5_{\pm 1.3}$ | 1.9 | 2.0 | $3.1_{\pm 0.9}$ | 6.4 |
| FR-RU | 0.3 | $2.1_{\pm 0.5}$ | - | $8.3_{\pm 4.4}$ | 4.5 | 12.5 | $11.1_{\pm 3.0}$ | 25.0 |
| FR-SV | 0.6 | $\mathbf{0.6_{\pm 0.0}}$ | 0.8 | $3.9_{\pm 1.4}$ | 2.7 | 2.1 | $5.1_{\pm 1.2}$ | 11.4 |
| IT-NL | 1.0 | $8.1_{\pm 0.4}$ | - | $4.6_{\pm 0.9}$ | **2.8** | 6.2 | $6.6_{\pm 0.8}$ | 8.9 |
| IT-PT | 0.4 | $\mathbf{0.4_{\pm 0.0}}$ | 0.5 | $2.2_{\pm 0.6}$ | 1.9 | 2.0 | $3.3_{\pm 0.9}$ | 5.9 |
| IT-RU | 0.6 | $4.9_{\pm 0.0}$ | - | $20.0_{\pm 15.1}$ | 8.8 | 27.2 | $26.9_{\pm 6.0}$ | 54.4 |
| IT-SV | 0.7 | $\mathbf{0.7_{\pm 0.0}}$ | 0.9 | $2.7_{\pm 0.2}$ | 2.7 | 1.2 | $4.5_{\pm 1.3}$ | 11.5 |

in the 2D and 3D cases. Although SHA is a competitive baseline, it often fails to reach the 99% threshold within the allocated budget. Notably, MO-ASHA, our strongest multi-objective baseline, is itself a multi-objective extension of SHA, underscoring the relevance of this comparison.

For the single-objective evaluation, all multi-fidelity baselines (SHA, Hyperband, and BOHB) are evaluated under an identical landmark fine-tuning schedule. This ensures that observed differences arise from the optimization strategies themselves rather than from discrepancies in budget allocation or fidelity control.

We emphasize that these methods are intentionally excluded from the main experiments, as they are inherently single-objective and rely on scalarized performance signals. Extending them to the multi-objective setting requires additional design choices (e.g., scalarization or dominance-based pruning) that can substantially influence performance and complicate interpretation. The single-objective ablation therefore serves to demonstrate that when these baselines are well-defined and evaluated under the same multi-fidelity protocol, LAMPS remains consistently superior.

# E. Datasets

This section describes the datasets used in our experiments.

### E.1. Text Classification

Although the datasets described here correspond to text classification tasks, they cover different NLP tasks, requiring different linguistic competencies, domain knowledge, and reasoning abilities. This diversity makes it particularly challenging (and well-suited) for evaluating LAMPS. For datasets without predefined training and validation splits, we reserve 20% of the data for validation.

**TREC**   A classic question classification benchmark with 6 coarse-grained classes (e.g., abbreviation, entity, description and abstract concept, human being, location, and numeric value). Task: Question classification. License: N/A (widely used academic benchmark; originally from UIUC).

**Clickbait**   Contains news headlines labeled as either "clickbait" or "non-clickbait". Derived from social media posts (Chakraborty et al., 2016). Task: Binary classification. License: N/A.

**20 Newsgroups**   A collection of $20,000$ newsgroup emails across 20 different topics (Lang, 1995). Task: Topic classification. License: CC BY 4.0.

**Amazon Reviews (cell-phone)**   Subset of the Amazon Product Review 2013 dataset, filtered for the "Cell Phone reviews" category. Includes star ratings from 1 to 5 and contains $78,930$ reviews. Task: Sentiment classification (5 classes). License: N/A (Amazon public data, widely used in academia).

**Hate Speech and Offensive Language**   A corpus of over 24,000 tweets manually annotated as hate speech, offensive but not hateful, or neither (Davidson et al., 2017). Task: Offensive language classification (3 classes). License: MIT License.

**MMLU**   Massive Multitask Language Understanding, a benchmark covering 57 diverse subject areas from elementary math to law and philosophy. Task: Multi-choice question answering. License: MIT License.

**Patent Classification**   Consisting of 35K Patent abstracts labeled with Cooperative Patent Classification (CPC) codes (9 classes). Task: Topic classification. License: Public domain (based on USPTO data).

**Emotion**   A dataset of 20K Twitter messages in English annotated with one of six basic emotions (anger, fear, joy, love, sadness, surprise). Task: Emotion classification. License: MIT License.

**CoLA**   Corpus of Linguistic Acceptability, a dataset of English sentences labeled as grammatically acceptable or unacceptable. Task: Acceptability classification (binary). License: Unknown (academic benchmark from the GLUE suite).

**SMS Spam**   A dataset of SMS messages labeled as spam or ham, widely used in spam detection research. Task: Binary classification. License: Open for research use.

**IMDB**   A large-scale movie review corpus containing 50K reviews labeled as positive or negative (Maas et al., 2011). Task: Sentiment classification (binary). License: Permissive research license.

**Financial Phrasebank**   A financial-domain sentiment dataset of short sentences annotated by multiple experts with high-agreement labels (positive, negative, neutral) (Malo et al., 2014). Task: Financial sentiment analysis (3 classes). License: Creative Commons Attribution-NonCommercial-ShareAlike 3.0 Unported License.

### E.2. Machine Translation

**OPUS Books**   A collection of copyright-free texts translated into multiple languages (Tiedemann, 2012). License: Available for personal, educational and research use.

## F. Pretrained Language Models

Below is the list of pretrained models used during the experiments of this paper:

**F.1. Text Classification**

**BERT:**

1. google-bert/bert-large-cased-whole-word-masking
2. google-bert/bert-large-uncased-whole-word-masking-fine-tuned-squad
3. google-bert/bert-large-uncased-whole-word-masking
4. google-bert/bert-large-uncased
5. google-bert/bert-large-cased-whole-word-masking-fine-tuned-squad
6. google-bert/bert-large-cased
7. google-bert/bert-base-uncased
8. google-bert/bert-base-multilingual-uncased
9. google-bert/bert-base-multilingual-cased
10. google-bert/bert-base-german-dbmdz-uncased
11. google-bert/bert-base-german-dbmdz-cased
12. google-bert/bert-base-german-cased
13. google-bert/bert-base-chinese
14. google-bert/bert-base-cased

**GPT:**

1. openai-community/gpt2
2. openai-community/gpt2-medium
3. openai-community/gpt2-large
4. openai-community/gpt2-xl

**RoBERTa:**

1. FacebookAI/roberta-base
2. FacebookAI/roberta-large
3. FacebookAI/xlm-roberta-base
4. FacebookAI/xlm-roberta-large
5. FacebookAI/xlm-roberta-large-fine-tuned-conll02-dutch
6. FacebookAI/xlm-roberta-large-fine-tuned-conll02-spanish
7. FacebookAI/xlm-roberta-large-fine-tuned-conll03-english
8. FacebookAI/xlm-roberta-large-fine-tuned-conll03-german

**OPT:**

1. facebook/opt-125m
2. facebook/opt-350m
3. facebook/opt-1.3b
4. facebook/opt-2.7b
5. facebook/opt-6.7b

**Llama:**

1. meta-llama/Llama-3.2-1B
2. meta-llama/Llama-3.2-1B-Instruct
3. meta-llama/Llama-3.2-3B
4. meta-llama/Llama-3.1-8B

**DistilBERT:**

1. distilbert/distilbert-base-multilingual-cased
2. distilbert/distilbert-base-german-cased
3. distilbert/distilbert-base-uncased-distilled-squad
4. distilbert/distilbert-base-cased-distilled-squad
5. distilbert/distilbert-base-cased
6. distilbert/distilbert-base-uncased
7. distilbert/distilroberta-base
8. distilbert/distilgpt2

**ALBERT:**

1. albert/albert-xlarge-v2
2. albert/albert-xxlarge-v2
3. albert/albert-xxlarge-v1
4. albert/albert-xlarge-v1
5. albert/albert-large-v2
6. albert/albert-large-v1
7. albert/albert-base-v2
8. albert/albert-base-v1

**LUKE:**

1. studio-ousia/mluke-large
2. studio-ousia/mluke-large-lite
3. studio-ousia/mluke-base-lite
4. studio-ousia/mluke-base
5. studio-ousia/luke-japanese-base
6. studio-ousia/luke-japanese-base-lite
7. studio-ousia/luke-japanese-large-lite
8. studio-ousia/luke-japanese-large
9. studio-ousia/luke-large-lite
10. studio-ousia/luke-base-lite
11. studio-ousia/luke-large
12. studio-ousia/luke-base

**DeepSeek:**

1. deepseek-ai/DeepSeek-R1-Distill-Qwen-1.5B
2. deepseek-ai/DeepSeek-R1-Distill-Qwen-7B
3. deepseek-ai/DeepSeek-R1-Distill-Llama-8B

**Qwen:**

1. Qwen/Qwen2.5-0.5B
2. Qwen/Qwen2.5-1.5B
3. Qwen/Qwen2.5-3B
4. Qwen/Qwen2.5-7B

**F.2. Machine Translation**

**Helsinki-NLP:**

1. Helsinki-NLP/opus-mt-en-sv
2. Helsinki-NLP/opus-mt-tc-bible-big-deu_eng_fra_por_spa-mul

**mBART:**

1. facebook/mbart-large-50
2. facebook/mbart-large-50-many-to-many-mmt
3. facebook/mbart-large-50-many-to-one-mmt
4. facebook/mbart-large-50-one-to-many-mmt
5. facebook/mbart-large-cc25
6. facebook/mbart-large-en-ro

**T5:**

1. google-t5/t5-3b
2. google-t5/t5-base
3. google-t5/t5-large
4. google-t5/t5-small
5. google/long-t5-local-large
6. google/long-t5-tglobal-xl

**mT5:**

1. google/mt5-base
2. google/mt5-large
3. google/mt5-small
4. google/mt5-xl

**UMT5:**

1. google/umt5-base
2. google/umt5-small

## G. Adding New Models to the Meta-Dataset

To incorporate a new model into the recommendation pool of LAMPS, it must first be integrated into the meta-dataset. We refer to this process as *model fingerprinting*. Because LAMPS relies on meta-learning, it is necessary to observe the actual performance of the new model on known datasets before the system can generalize its behavior to unseen datasets. This integration requires two steps:

1. The new LLM must be fine-tuned on all datasets currently included in the meta-dataset, with all relevant metrics recorded.

2. The reinforcement learning policy must be retrained on the expanded meta-dataset.

Currently, complete retraining is the recommended procedure for reliable integration of new models. Although incremental training strategies could further reduce the computational overhead, the cost of full retraining is already negligible compared to the fine-tuning runs required to expand the meta-dataset.

The ideal number and diversity of datasets in the meta-dataset remains an open research question. A smaller set of datasets facilitates the addition of new models, since each integration requires fewer fine-tuning runs. Conversely, a larger and more diverse collection typically improves the generalization ability of the learned policy to unseen tasks. How to balance these competing goals remains an open challenge for future work.

