# OpenReview forum: "Landmark-Guided Policy Optimization for Multi-Objective Language Model Selection"
_ICML.cc/2026/Conference — ICML 2026 regular_

### Official Review · Reviewer_SSKU · 2026-02-23

**Soundness:** 2
**Presentation:** 3
**Significance:** 2
**Originality:** 3
**Overall Recommendation:** 3
**Confidence:** 3

**Summary:**

This paper proposes LAMPS, a multi-objective AutoML framework that identifies the Pareto front of candidate LLMs via a meta-learned resource allocation policy. Two core components: landmark fine-tuning, which partitions training data into exponentially growing subsets to generate early learning curve signals, and multi-task RL based on PPO.

**Compliance With Llm Reviewing Policy:**

Affirmed.

**Final Justification:**

The rebuttal clarified the relationship to Quick-Tune and DyHPO but did not resolve my core concerns. Model size is static and known before fine-tuning, so the multi-objective structure largely degenerates to per-group single-objective selection, and the practical motivation for generating a full Pareto front is unclear when deployment constraints are typically known in advance. The necessity of RL over BO was argued conceptually but not established empirically. I prefer to maintain my score.

**Key Questions For Authors:**

Q1. [1] combines meta-learning, learning curve signals, and model selection, extended to LLM fine-tuning. What are the technical differences, and what advantage does RL offer over BO in the multi-objective setting?

Q2. Model size is a static property known before fine-tuning, which substantially simplifies the multi-objective structure. If replaced with a dynamic objective such as inference latency or training time, how does LAMPS perform?

**Strengths And Weaknesses:**

Strengths

1. The problem formulation is clear and the multi-objective framing is reasonable. Quantifying exhaustive search cost in dollar terms, $5,141.67 for Amazon, $449.58 for LAMPS, strengthens the practical motivation.
2. The combination of landmark fine-tuning and RL meta-learning is principled: early learning curve signals predicting final performance is well-supported in the literature, and multi-task RL enables zero-shot generalization to new datasets.

Weaknesses

1. The most directly related prior work is missing. [1] combines meta-learning with gray-box HPO, learning curve signals, and pretrained model selection, later extended to LLM fine-tuning. The core difference is only the choice of BO vs. RL and single-objective vs. multi-objective scope. Omitting both discussion and comparison is a serious gap. [2] similarly uses learning-curve-aware dynamic resource allocation and is also not cited.

2. The multi-objective problem setup does not match practical deployment scenarios. Practitioners face two situations: when a hard constraint exists, such as a maximum parameter count or VRAM limit, the problem reduces to finding the lowest validation loss among models satisfying the constraint, which is constrained single-objective optimization requiring no Pareto front. Without a constraint, one simply picks the model with the lowest validation loss. The paper's premise of "generate a Pareto front and let users choose a tradeoff later" assumes constraints are unknown at search time, which is rarely the case. More fundamentally, model size, the paper's second objective, is a static property known before any fine-tuning begins. The Pareto front therefore degenerates to: for each discrete parameter count level, find the model with the lowest post-fine-tuning validation loss. This is per-group single-objective selection, not genuine multi-objective optimization.

3. The necessity of RL is not established. The agent's observation is each model's current validation loss, model size, and epochs trained; the action is selecting which model to train next. Training model A for one epoch has no effect on subsequently training model B, so there is no genuine temporal dependency between actions. The problem is fundamentally one of predicting which models will be Pareto-optimal from partial learning curves, a supervised prediction problem rather than a sequential decision problem. [2] addresses exactly this setting using Bayesian optimization to update performance predictions after each observation and select the next candidate, which is the natural baseline.

[1] Arango, Sebastian Pineda, et al. "Quick-tune: Quickly learning which pretrained model to finetune and how." arXiv preprint arXiv:2306.03828 (2023).

[2] Wistuba, Martin, Arlind Kadra, and Josif Grabocka. "Supervising the multi-fidelity race of hyperparameter configurations." Advances in Neural Information Processing Systems 35 (2022): 13470-13484.

---

> ### Author Rebuttal · Authors · 2026-03-31
>
> We thank the reviewer for the constructive feedback and thoughtful questions. We appreciate the opportunity to clarify the positioning of our work and to better explain the design choices behind LAMPS.
>
> > Quick-tune & DyHPO
>
> We will add a discussion of both references in the camera-ready version. Quick-Tune addresses a different problem setting than our LAMPS. Quick-Tune is a single-objective CASH-style method, originally developed for image classification; its later LLM extension [3] fine-tunes a single backbone model (Microsoft Phi-3). Similar, DyHPO is designed for surrogate-based single-objective optimization. In contrast, LAMPS performs multi-model Pareto selection. Therefore, a direct empirical comparison is not appropriate.
>
> [3] T. Strangmann et al. "Transfer Learning for Finetuning Large Language Models." Accepted at the NeurIPS 2024 Workshop on Adaptive Foundation Models.
>
> > Differences and advantages of RL over BO in the multi-objective setting
>
> BO is a natural alternative here, and we do not claim that RL is inherently superior. Our motivation for using RL is narrower: LAMPS aims to learn a **resource-allocation policy** that directly maximizes Pareto-front recovery from partial trajectories. BO could also be extended to this setting, but it would require designing a Pareto-aware multi-fidelity surrogate and acquisition rule. RL instead lets us optimize the allocation policy end-to-end from replayed meta-dataset trajectories, without introducing an intermediate predictive model. We therefore view RL and BO as complementary approaches, with RL being a natural fit for the objective studied here; incorporating BO-style signals into the RL setup is a promising direction for future work.
>
> > Model size vs. dynamic objective
>
> Our initial experiments for LAMPS used **training time** and validation loss as the primary objectives, and the initial results were similar. However, training time is strongly dependent on the hardware setup, making comparisons less reproducible across environments, and the resulting meta-dataset harder to reuse. For this reason, we replaced training time with model size, which serves as a more stable and portable proxy for practical deployment cost.

---

> > ### Author Rebuttal · Reviewer_SSKU · 2026-04-02
> >
> > Thank you for the rebuttal. My core concerns remain unresolved. The multi-objective formulation is questionable given that model size is a static property, and the practical motivation is unclear when deployment constraints are known in advance. The necessity of RL over BO is also not empirically established. Based on this, I prefer to maintain my score.

---

> > > ### Author Response · Authors · 2026-04-08
> > >
> > > We thank you again for the thoughtful follow-up. We agree that, when deployment constraints are known and fixed in advance, the problem can often be posed as a constrained single-objective. We do not view this, however, as the only practically relevant regime.
> > >
> > > In many real deployments, requirements may change with system conditions, budget availability, or business priorities. For example, during peak hours an administrator may prefer a cheaper model while tolerating a small loss in performance. In such cases, recovering the Pareto front in a single run is useful because it makes the cost-performance trade-offs explicit, enabling more informed decisions without rerunning the search whenever requirements change, as would typically be necessary under a single-objective formulation tied to fixed deployment requirements or preferences.
> > >
> > > Therefore, we understand the remaining disagreement as being primarily about the modeling choice / deployment regime, rather than about the technical validity of the method itself. That said, we will make this positioning clearer in the camera-ready version, including a more explicit statement that constrained single-objective search is a reasonable alternative when requirements are fixed in advance, whereas LAMPS is targeted at settings where those requirements may evolve over time.

---

### Official Review · Reviewer_6puj · 2026-03-09

**Soundness:** 2
**Presentation:** 3
**Significance:** 2
**Originality:** 3
**Overall Recommendation:** 4
**Confidence:** 3

**Summary:**

This paper studies how to choose pretrained language models for downstream fine-tuning when there are multiple goals, such as getting low validation loss while keeping the model small. Instead of selecting a single best model, it tries to find a good Pareto set of trade-offs. The proposed method, LAMPS, first uses landmark fine-tuning to get early signals by training models on gradually larger subsets of the training data, and then uses a learned PPO policy to decide which model should receive the next unit of training budget. On a new dataset, the policy keeps allocating budget across candidates until the search budget is used up, and then removes dominated models. Experiments on 70 pretrained LLMs and 12 held-out datasets show that LAMPS can reach 99% of the optimal hypervolume with 73.6% less search time than exhaustive search on average, and it performs better than the compared baselines on 9 of 12 datasets.

**Compliance With Llm Reviewing Policy:**

Affirmed.

**Final Justification:**

The paper studies a practical and meaningful problem, namely multi-objective base-model selection for downstream fine-tuning, and proposes a coherent framework combining landmark fine-tuning with a meta-learned allocation policy. The empirical results suggest that the method can substantially reduce search cost while still recovering strong Pareto solutions.

The rebuttal improved my assessment. In particular, the authors clarified the distinction between deployment-time search cost and the one-time offline meta-dataset construction cost, and added more concrete discussion of the practical overhead. These additions strengthen the empirical case, and I have raised my score accordingly.

Some concerns remain only partially resolved, especially around the sparse terminal reward design, policy-training stability, and the sensitivity and role of landmark fine-tuning. Still, the rebuttal makes the overall contribution more convincing, and I view the paper more positively than before.

**Key Questions For Authors:**

Does the search time in Figure 3 include the cost of meta-dataset collection and PPO training, or only test-time search on the held-out dataset? If this cost is excluded and is specific to LAMPS, its practical overhead should be discussed more explicitly.

**Limitations:**

The main limitations of the paper are that its empirical setting is still somewhat idealized and its deployment assumptions may be optimistic. In particular, the primary objectives are validation loss and model size rather than fully measured deployment metrics; the policy is trained with privileged termination information unavailable at test time; and the overall framework depends on a preconstructed meta-dataset whose expansion requires costly model fingerprinting and retraining. As a result, while the method is promising as a meta-learned search framework, its real-world scalability and robustness to rapidly evolving model pools are not yet fully established.

**Strengths And Weaknesses:**

## Strengths

- **The problem is practically meaningful and well framed.**  The paper treats base-model selection as a multi-objective search problem rather than a single-metric ranking problem, which is a more realistic formulation for downstream deployment settings. The use of Pareto optimality and hypervolume is also standard and principled.
- **The method is conceptually clean.**  The combination of landmark fine-tuning and sequential resource allocation is intuitive: early partial learning curves provide informative signals, and the RL policy uses them to decide where to spend additional fine-tuning budget. This makes the method easy to follow and reasonably well motivated.
- **The empirical gains are meaningful.**  The reported speedup over exhaustive search is large, and the method is competitive against MO-ASHA and other baselines on most held-out datasets.

## Weaknesses

- **The evaluation is somewhat aggregate-heavy.** Figure 3 mainly reports the search time to reach 99% of the optimal hypervolume, and Figure 4 shows the average hypervolume-loss curves, but the paper provides little qualitative analysis of what each method actually selects. Since the goal is to recover a near-Pareto set, it would be helpful to include at least one case study showing the selected models, the recovered Pareto front, and its gap to the exhaustive-search Pareto front on a held-out dataset.
- **The reward design raises some questions.** The policy is trained with a sparse terminal reward, with zero reward at all intermediate steps. While the appendix claims that this reshaping improves PPO stability, the paper does not report basic training diagnostics such as the average episode length, reward variance across seeds, or comparisons against denser reward alternatives. Given the long-horizon sequential allocation setup, it is unclear how stable and sample-efficient policy learning is in practice.
- **The role of landmark fine-tuning is not fully characterized.** Although the ablation in Appendix E shows that removing landmarks slows convergence and lowers the final reward, this only establishes that landmarks are useful in aggregate. The paper does not analyze how sensitive LAMPS is to the number of landmarks $K$, or the extent to which the gains come from landmarking itself versus the policy learner built on top of it.
- **The paper lacks a failure-case analysis.** Although LAMPS wins on 9 of 12 datasets, MO-ASHA remains better on 3 datasets, and the paper does not analyze what distinguishes these cases. A more informative discussion of when the meta-learned policy helps, and when a strong generic baseline like MO-ASHA is preferable, would make the empirical claims much stronger.

Overall, I think this is a good paper, but the experiments are currently not concrete enough, and I would be open to raising my score if the empirical analysis is strengthened.

---

> ### Author Rebuttal · Authors · 2026-03-31
>
> Thank you for the careful review and helpful suggestions, which we have addressed as detailed below.
>
> Figure 3 reports only the deployment-time search cost on the held-out dataset, not the one-time offline cost of meta-dataset construction and PPO training. The dominant overhead is meta-dataset collection, while PPO training is negligible in comparison (as discussed in Appendix H). As quantified in our response to **Reviewer G9xv**, this upfront cost is 5,747 GPU-hours, which is amortized, on average, after 13.9 future tasks. We therefore view the meta-dataset as a shared offline resource: built once and then reused broadly.
>
> To make the behavior more concrete, we inspected individual search trajectories. On [Amazon (Cell Phone)](https://ibb.co/tTTfFjPf), LAMPS demonstrates an efficient strategy: first explores inexpensive ALBERT variants, then shifts budget to stronger candidates such as Qwen2.5-3B, reaching its first substantial HV improvement at ~90 GPU-hours and refining the Pareto set thereafter. On [Hate Speech](https://ibb.co/DDWZsPYf), in contrast, LAMPS spends many allocations on initially promising candidates, notably Llama-3.1-8B, yet the HV remains nearly flat for a long stretch and improves substantially only after later switches. This failure mode is consistent with the RU translation analysis discussed in our response to **Reviewer G9xv**: LAMPS is strongest when the target task is well represented in the meta-dataset, while more isolated tasks are harder to transfer to.
>
> Finally, also in our responses to **YcNW** and **G9xv**, we added further empirical analyses on budget sensitivity, LoRA compatibility, and partial integration of new models, which may also be of interest here. We hope these additions make the practical overhead, search behavior, and failure modes of LAMPS more concrete.

---

> > ### Author Rebuttal · Reviewer_6puj · 2026-04-02
> >
> > Thank you for the detailed rebuttal and for adding several concrete empirical analyses. I appreciate that the response clarifies the practical overhead of LAMPS more explicitly, including the separation between deployment-time search cost and the one-time offline meta-dataset construction cost, as well as the break-even discussion. These results make the practical behavior of the method more concrete and improve my confidence in the paper. Accordingly, I have raised my score.
> >
> > That said, some of my concerns are only partially resolved. In particular, I still think the paper would benefit from a clearer analysis of the sparse terminal reward design and policy-training stability, as well as a more direct characterization of the role and sensitivity of landmark fine-tuning. So while the rebuttal does not fully remove all of my reservations, it does substantially strengthen the empirical case for the paper and makes the overall contribution more convincing.

---

> > > ### Author Response · Authors · 2026-04-08
> > >
> > > We thank you for the thoughtful follow-up and appreciate that our rebuttal strengthened the empirical case for the paper, made the overall contribution more convincing, and led to a more positive assessment of our work. While some minor concerns remain, it is good to see that they did not outweigh your strengthened overall evaluation, and we will discuss them in the camera-ready version.

---

### Official Review · Reviewer_YcNW · 2026-03-12

**Soundness:** 3
**Presentation:** 2
**Significance:** 3
**Originality:** 2
**Overall Recommendation:** 4
**Confidence:** 3

**Summary:**

LAMPS is a framework that tackles the high computational cost of selecting a pretrained LLM for downstream fine-tuning by formulating it as a multi-objective optimization problem. It integrates landmark fine-tuning, which gathers early performance signals by evaluating models on incrementally growing data subsets, with a meta-learned reinforcement learning policy trained via PPO on historical learning trajectories. This policy manages resource allocation, learning to prioritize promising models and early-stop poorly performing candidates. Across tasks like text classification and machine translation, LAMPS reduces the search time compared to an exhaustive search while preserving the target space hypervolume on unseen datasets.

**Compliance With Llm Reviewing Policy:**

Affirmed.

**Final Justification:**

Authors addressed my concerns via added experiments. I have increase my significance score accordingly and maintain my positive evaluation of the paper.

**Key Questions For Authors:**

Please see above

**Limitations:**

Please see above

**Strengths And Weaknesses:**

> Strengths

1. The formulation of LLM selection accounts for practical deployment trade-offs like model size and inference costs, moving beyond the standard single-objective focus on test loss.
2. Integrating subsampling landmarks with multi-task reinforcement learning efficiently translates early and partial learning curves into actionable decisions that may generalize to unseen datasets.

> Weaknesses

1. Heavy Reliance on Full-Model Fine-Tuning

The framework evaluates candidate models using full-parameter fine-tuning, which requires massive computational overhead to build the meta-dataset and limits applicability to PEFT setups.

Thus, can authors Implement LoRA on a subset of the text classification tasks (e.g., TREC) to verify if the early landmark signals generated by parameter-efficient updates remain robustly predictive of final model performance?

2. High Cost of Integrating New Models

Adding a new candidate model to the selection pool requires an expensive "fingerprinting" process where the new model must be fine-tuned on all historical datasets, followed by a full retraining of the RL policy.

Can authors perform a zero-shot integration test by holding out one existing LLM (e.g., a specific Llama variant) from the meta-dataset, fingerprinting it on only 10% of the historical datasets, and evaluating if the PPO agent can still accurately allocate resources to it during inference?

---

> ### Author Rebuttal · Authors · 2026-03-31
>
> Dear reviewer,
>
> We sincerely appreciate the thoughtful questions. To address them directly, we conducted additional experiments and summarize the results below.
>
> > LoRA loss curves
>
> As requested, we compared LoRA and full fine-tuning on several candidate models on TREC. Although LoRA typically yields [higher validation loss](https://ibb.co/Q7rwh4pt), the shape of the curves follows the same pattern as observed in Figure 2, and the [accuracy trajectories](https://ibb.co/C32dgWNM) follow a very similar progression. This suggests that the landmark signals remain informative under LoRA, even if their absolute loss values differ. LoRA also substantially reduces runtime, making it a promising PEFT alternative in this setting.
>
> > Integration of new models with incomplete data in the meta-dataset
>
> To evaluate this, we considered the held-out Amazon (Cell Phone) dataset and Qwen2.5-7B, which is Pareto-optimal on this task. We then removed this model from policy training on N randomly chosen historical datasets (5 seeds) and measured how much performance degrades when the policy is evaluated with an incomplete fingerprint for that model. The results are shown below.
>
> | N removed | GPU-hours to reach 99% HV |
> |----------------|-----------------------------------------|
> | 0  | 122.5 ± 47.1 (from Fig. 3) |
> | 2  | 134.9 ± 36.4               |
> | 6  | 141.1 ± 69.5               |
> | 10 | 168.3 ± 50.2               |
>
> As expected, performance degrades as more historical fingerprints are removed. Still, even in the most extreme setting we tested (N = 10, i.e., retaining only ~10% of the historical fingerprints), LAMPS remains better than MO-ASHA on this dataset (230.3 ± 47.1 GPU-hours to reach 99% HV). This suggests that integrating a new model may not always require full fingerprinting on all historical datasets: partial fingerprinting can still preserve a substantial fraction of LAMPS’s benefit.

---

> > ### Author Rebuttal · Reviewer_YcNW · 2026-04-01
> >
> > Thank you for addressing my questions with the added empirical results. I would like to maintain my positive evaluation of the paper.

---

> > > ### Author Response · Authors · 2026-04-08
> > >
> > > We are glad that our rebuttal and added empirical results adequately addressed your questions, and we appreciate your positive assessment of the paper.

---

### Official Review · Reviewer_G9xv · 2026-03-13

**Soundness:** 3
**Presentation:** 3
**Significance:** 3
**Originality:** 3
**Overall Recommendation:** 4
**Confidence:** 3

**Summary:**

This paper introduces LAMPS, a multi-objective AutoML framework that meta-learns a scheduling policy for efficiently recovering the Pareto front of candidate pretrained LLMs for fine-tuning. The method combines landmark fine-tuning (evaluating models on exponentially growing data subsets for early multi-fidelity signals) with a PPO-based policy trained on historical trajectories of 70 pretrained LLMs. The optimization maximizes the hypervolume indicator with a time-aware terminal reward whose optimality-preserving property is formally established (Lemma D.1). Evaluated via leave-one-out cross-validation on 12 text classification datasets (5 seeds) and 38 machine translation directions (2- and 3-objective settings), LAMPS reduces search time by 73.6% versus exhaustive search while maintaining over 99% of optimal hypervolume.

**Compliance With Llm Reviewing Policy:**

Affirmed.

**Final Justification:**

The rebuttal has strengthened the paper on all four questions I raised. The budget-sensitivity result (Q2) is particularly strong and removes my primary concern about training-deployment mismatch. The LogME comparison (Q3) validates the RL mechanism's contribution. The break-even analysis (Q1) and LoRA/failure-mode results (Q4) are acceptable though not fully comprehensive. I maintain my overall score of 4 (Weak Accept) and now hold this position with increased confidence. The paper presents a principled framework addressing a practically important problem, with strong experimental results and responsible disclosure. The rebuttal has adequately contextualized the meta-dataset cost and demonstrated that the efficiency gains are robust under realistic deployment constraints. I support acceptance.

**Key Questions For Authors:**

1. **Meta-dataset amortization (critical).** How many GPU-hours were required to construct the meta-dataset, and after how many new tasks does LAMPS break even versus exhaustive search? *A favorable result (5–10 tasks) would directly strengthen significance; an unfavorable one (50+) would substantially reduce practical impact.*

2. **Budget-sensitivity analysis (critical).** Can you provide hypervolume-vs-budget curves showing how coverage degrades under tighter deployment budgets? *If coverage remains above 95% at 50% budget, the training-deployment gap concern (W2) is largely resolved.*

3. **LogME comparison (critical).** Can you compare against a two-stage baseline using LogME to pre-filter candidates before MO-ASHA? *This would calibrate how much the RL machinery contributes beyond the landmark schedule's inductive bias.*

4. **Russian failure analysis and LoRA compatibility.** What explains the consistent underperformance on RU-involving pairs? Can LAMPS transfer to LoRA-based fine-tuning without re-collecting the meta-dataset? *Both would materially improve the generalization and practical applicability narrative.*

**Limitations:**

Yes

**Strengths And Weaknesses:**

### **Strengths**

**1. Principled formulation of a practically important problem.** The paper correctly identifies multi-objective LLM selection as a timely deployment need and proposes a coherent framework where landmark fine-tuning (early signals) and meta-learned PPO scheduling (resource allocation) naturally reinforce each other. The reward design is theoretically grounded via Lemma D.1, which formally proves optimality preservation—uncommon rigor for RL-for-AutoML work.

**2. Impressive experimental scale with strong results.** The meta-dataset spans 70 models (ALBERT ~11M to DeepSeek-R1 ~8B) across diverse architectures. Leave-one-out evaluation with 5 seeds reports standard deviations, and the extension to 38 translation directions and 3-objective settings demonstrates scalability. LAMPS wins on 9/12 classification tasks and 35/38 single-objective comparisons, with a concrete Amazon cost analysis ($5,141 → $449) illustrating economic impact.

**3. Responsible disclosure and reproducibility.** The appendices provide comprehensive hyperparameters, model inventories, and setup details. The Impact Statement substantively discusses both positive effects (energy savings, accessibility) and risks (accelerated deployment without adequate evaluation, omitted fairness objectives), exceeding the venue's typical standard.

---

### **Weaknesses**

**1. Unquantified upfront cost undermines the efficiency narrative.** LAMPS requires *complete* fine-tuning trajectories for all 70 models across all meta-training datasets—an enormous investment that is never quantified. Without a break-even analysis, the 73% time savings cannot be contextualized. Moreover, adding a new model requires fine-tuning it on all existing tasks and retraining the policy (Appendix H), limiting practicality in rapidly evolving model ecosystems. This is the paper's most significant practical limitation.

**2. Training-deployment gap and systematic failure modes are unanalyzed.** Training episodes terminate only when all Pareto-optimal models are fully fine-tuned (privileged information), while deployment uses fixed budgets. No budget-sensitivity analysis quantifies how policy quality degrades under tighter budgets. Additionally, appendix tables reveal consistent underperformance on Russian-involving language pairs (3–5× slower than Oracle) and 3-objective degradation (e.g., EN-FR: 222.4 vs. Oracle 77.6), but no failure diagnosis is provided. The observation space—limited to current objectives and completed epochs, without model metadata or dataset features—likely contributes to these generalization issues.

**3. Insufficient baselines and ablations narrow the evidence base.** Only MO-ASHA serves as a genuine multi-objective competitor; model selection methods (LogME) and transferable HPO methods (warmstarted BOHB, ParEGO) discussed in Related Work are not compared experimentally. Ablations cover only landmark removal; the number of landmarks $K$, reward shaping contribution, invalid action masking, and the unusually high entropy coefficient ($c_2=0.23$) remain unisolated. The restriction to full fine-tuning (no LoRA/PEFT) and narrow task coverage (classification + translation only) further limit scope relative to the paper's framing.

---

> ### Author Rebuttal · Authors · 2026-03-31
>
> Dear reviewer,
>
> We sincerely appreciate the thoughtful questions. We performed additional experiments to address them as directly as possible. We answer each point concisely below and are glad to provide further details during the discussion phase.
>
> > Break-even point
>
> The meta-dataset construction cost (70 models, 12 datasets) was 5,747 GPU-hours. Under the leave-one-out protocol, this corresponds to a mean upfront cost of 5,268 GPU-hours for 11 meta-training datasets. Since LAMPS saves 378.8 GPU-hours per new task on average, this upfront cost is amortized after **13.9 future tasks**.
>
> > Hypervolume vs budget
>
> From Figure 4, taking the exhaustive-search GPU-hours as 100% budget, LAMPS reaches 95% HV coverage using only **15.23%** of that budget on average, well below the 50% budget level raised in the reviewer's question.
>
> > LogME + MO-ASHA
>
> We compared against a two-stage baseline on TREC, first using LogME to retain the [top 30% of candidate models](https://ibb.co/NnQdmHgb) via non-dominated sorting, and then run MO-ASHA on the retained set. As shown below, this baseline improves slightly over vanilla MO-ASHA at very early budgets, but it also filters out several truly Pareto-optimal models and, therefore, never reaches 99% HV coverage ($\log_{10}(1-0.99) = -2.0$ HV loss). This suggests that a static pre-filter can help early exploration, but is not sufficient for our multi-objective setting.
>
> Table: Evolution of hypervolume loss on TREC heldout-dataset.
>
> | t    | MO-ASHA | LogME + MO-ASHA | LAMPS   |
> |------|---------|------------------|---------|
> | 20h  | -1.7471 | -1.8097          | -0.9270 |
> | 40h  | -1.8570 | -1.9431          | -1.4935 |
> | 60h  | -1.8665 | -1.9508          | -2.2076 |
> | 80h  | -1.8861 | N/A              | -3.5229 |
> | 100h | -3.5229 | N/A              | -4.0000 |
>
>
> > Failure-mode
>
> Regarding the RU-related failure mode, we empirically measured pairwise dataset similarity by training the policy on a single dataset and evaluating it on all others, with higher transfer reward indicating greater source-target similarity. A [2D t-SNE visualization](https://ibb.co/0p4cY5GM) of this matrix shows that the Russian translation datasets form a tight cluster that is highly isolated from the rest. We also computed each dataset’s mean distance to all others; the top outliers were: es-ru, de-ru, en-ru, it-ru, fr-ru, en-fi, de-en, fi-fr. This aligns closely with the datasets on which LAMPS underperforms.
>
> This suggests that the current policy covers the majority well, but transfers less effectively to marginal datasets. Improving such edge cases would likely require filtering or reweighting the meta-training datasets and training a new policy specifically for the target dataset, reducing negative transfer from unrelated datasets. We appreciate this helpful suggestion.
>
> > LoRA on test time only
>
> We experimented with LoRA at test time on TREC, while keeping the policy and meta-dataset unchanged, and using LoRA only for models larger than 3B parameters (e.g., DeepSeek, Llama, OPT, Qwen). This reduced search time by ~44%, while yielding nearly the same downstream accuracy as full fine-tuning for those models. We also did not observe any unstable behavior: the [landmark loss curves](https://ibb.co/Q7rwh4pt) followed the same overall shape as in Figure 2, differing mainly in their initial values. While preliminary, this suggests that LAMPS can transfer to a LoRA-based test-time regime without re-collecting the meta-dataset. We thank the reviewer for this suggestion.

---

> > ### Author Rebuttal · Reviewer_G9xv · 2026-04-03
> >
> > The rebuttal has strengthened the paper on all four questions I raised. The budget-sensitivity result (Q2) is particularly strong and removes my primary concern about training-deployment mismatch. The LogME comparison (Q3) validates the RL mechanism's contribution. The break-even analysis (Q1) and LoRA/failure-mode results (Q4) are acceptable though not fully comprehensive. I maintain my overall score of **4 (Weak Accept)** and now hold this position with increased confidence. The paper presents a principled framework addressing a practically important problem, with strong experimental results and responsible disclosure. The rebuttal has adequately contextualized the meta-dataset cost and demonstrated that the efficiency gains are robust under realistic deployment constraints. I support acceptance.

---

> > > ### Author Response · Authors · 2026-04-08
> > >
> > > We are glad that our rebuttal fully addressed your concerns and increased your confidence in our work, and we appreciate your explicit support for acceptance.

---

### Decision · Program_Chairs · 2026-04-30

**Decision:**

Accept (regular)

**Comment:**

This paper presents LAMPS, a multi‑objective AutoML framework for selecting pretrained LLMs for downstream fine‑tuning. The method combines landmark fine‑tuning to obtain early performance signals with a meta‑learned PPO scheduling policy that allocates fine‑tuning budget to efficiently recover the Pareto front. Across large‑scale experiments on text classification and machine translation, LAMPS substantially reduces search cost while maintaining near‑optimal hypervolume.

Reviewers consistently found the problem formulation meaningful, the framework principled, and the experimental scale impressive. Strengths highlighted include the clear multi‑objective framing, strong empirical results and thorough experimental reporting. \

The main concerns focused on practical overhead and modeling assumptions. Other raised questions were about baseline coverage, ablation completeness, failure cases on certain datasets, and whether RL is strictly necessary compared to alternatives such as BO.
The rebuttal addressed most major concerns by providing a break‑even analysis, budget‑sensitivity results, additional baselines (including LogME filtering), experiments on LoRA compatibility, and analysis of failure modes and partial model fingerprinting.

Overall, I'd rate this paper as borderline, leaning towards a weak accept given that overall, LAMPS is a technically solid and well‑evaluated contribution to multi‑objective LLM selection, even though some questions and concerns about generality and deployment assumptions remain.